

# Unrepresented model errors - effect on estimated soil hydraulic material properties

Stefan Jaumann[1,2] and Kurt Roth[1,3]

[1]Institute of Environmental Physics, Heidelberg University, Im Neuenheimer Feld 229, 69120 Heidelberg, Germany
[2]HGSMathComp, Heidelberg University, Im Neuenheimer Feld 205, 69120 Heidelberg, Germany
[3]Interdisciplinary Center for Scientific Computing, Heidelberg University, Im Neuenheimer Feld 205, 69120 Heidelberg, Germany

*Correspondence to:* S. Jaumann (stefan.jaumann@iup.uni-heidelberg.de)

**Abstract.** We investigate the quantitative effect of unrepresented (i) sensor position uncertainty, (ii) small scale-heterogeneity, and (iii) 2D flow phenomena on estimated effective soil hydraulic material properties.

Therefore, a complicated 2D subsurface architecture (ASSESS) was forced with a fluctuating groundwater table. Time Domain Reflectometry (TDR), Ground Penetrating Radar (GPR), and hydraulic potential measurement devices monitored the hydraulic state during the experiment. Since the measurement data are analyzed with an inversion method, starting close to the measurement data is key. Therefore, we developed a method which estimates the initial water content distribution by approximating the soil water characteristic on the basis of TDR measurement data and the position of the groundwater table. In order to reduce parameter bias due to unrepresented model errors, we implemented a structural error analysis to identify uncertain model components which have to be included in the parameter estimation. Hence, focussing on TDR and hydraulic potential data, we realized a 1D and a 2D study with increasingly complex setups: Starting with estimating effective hydraulic material properties, we added the estimation of sensor positions, the estimation of small-scale heterogeneity, or both.

The analysis of these studies with a modified Levenberg-Marquardt algorithm demonstrates three main points: (i) The approximated soil water characteristic for the initial water content distribution is reasonably close to inversion results. (ii) Although the material properties resulting from 1D and 2D studies are similar, the 1D studies are likely to yield biased parameters due to unrepresented lateral flow. (iii) Representing and estimating sensor positions as well as small-scale heterogeneity improves the mean absolute error by more than a factor of 2.

## 1 Introduction

Soil hydraulic material properties are essential to advance quantitative understanding of soil water dynamics. Despite decades of research, direct identification of these properties is expensive and near to impossible at larger scales. Therefore, indirect identification methods, such as inversion methods (Hopmans and Šimůnek, 1999; Vrugt et al., 2008), have been successfully applied to evaluate many experiments starting from lab-scale with One-Step Outflow (Parker et al., 1985), Multistep Outflow (Van Dam et al., 1994), and evaporation (Šimůnek et al., 1998; Schneider et al., 2006), up to field scale studies (Wollschläger et al., 2009; Huisman et al., 2010). Due to the multi-scale heterogeneity of the soil hydraulic material properties (Nielsen





et al., 1973; Gelhar, 1986; Cushman, 1990; Vogel and Roth, 2003), effective material properties have to be identified directly at the scale of interest. Yet, most studies focus on 1D subsurface architectures with homogeneous layers, e.g., Abbaspour et al. (2000); Ritter et al. (2003); Mertens et al. (2006); Wöhling et al. (2008); Wollschläger et al. (2009). Only few studies, e.g., Abbasi et al. (2004); Palla et al. (2009); Huisman et al. (2010), estimate material properties of effectively 2D subsurface

architectures. Abbasi et al. (2004) conducted an irrigation experiment to estimate soil hydraulic and solute transport properties for a 2D furrow architecture. Based on subsurface flow hydrographs for eight rain events, Palla et al. (2009) estimated effective soil hydraulic properties for a 2D layered coarse grained green roof. Exploiting flat wire Time Domain Reflectometry (TDR) and electrical resistance tomography (ERT) measurement data recorded during a fluctuating groundwater table experiment, Huisman et al. (2010) estimated soil hydraulic properties of a homogeneous dike.

Being common practice, these studies neglect critical uncertainties, e.g., concerning the input error or small-scale heterogeneity, and restrict the number of estimated material parameters to a minimal amount. Our main hypothesis is that this procedure leads to biased estimates for effective soil hydraulic properties due to neglected processes.

We show for a 1D and a 2D study, that representing and estimating uncertain model components improves the quality of the representation significantly (Sect. 5). These studies are setup according to an uncertainty analysis indicating which uncertainties

to represent (Sect. 4). Providing the measurement data (Sect. 3) for this analysis, Time Domain Reflectometry (TDR), Ground Penetrating Radar (GPR), and hydraulic potential measurement devices monitored the hydraulic system (Sect. 2) while it was forced with a fluctuating groundwater table.

## 2   ASSESS

Our test site *ASSESS* is located near Heidelberg, Germany, and consists of three different kinds of sand (material A, B, and

C) with different grain size distributions (Table 1). Its effective 2D subsurface architecture is visualized in Fig. 1. The approximately $2\,\mathrm{m} \times 20\,\mathrm{m} \times 4\,\mathrm{m}$ large site is equipped with a weatherstation[1], 32 TDR sensors[2], one tensiometer (UMS T4-191), and a well to monitor and manipulate the groundwater table. A geotextile separates the sand from an approximately $0.1\,\mathrm{m}$ thick gravel layer below, which ensures a rapid water pressure distribution and is the only connection of the groundwater well with the rest of the test site. Below this gravel layer, a concrete layer confines the site. As the test site is built into a former

fodder-silo, a concrete L-element serves as additional wall. In order to stabilize the material during the construction, it was compacted. Beyond the compaction interfaces shown in Fig. 1, GPR measurements, e.g., presented by Klenk et al. (2015a), indicate even more compaction interfaces.

We use this site to improve and develop GPR measurement and evaluation methods which increase the quantitative understanding of soil water dynamics. These methods comprise water content measurement (Buchner et al., 2011), estimation of the position of material interfaces as well as the effective relative permittivity distribution (Buchner et al., 2012), identification

---

[1]The weatherstation measures precipitation, relative humidity, radiation, wind direction, and wind velocity.
[2]Each TDR sensor has three rods (length: 0.20 m, diameter: 0.005 m) and is associated with a soil temperature sensor.





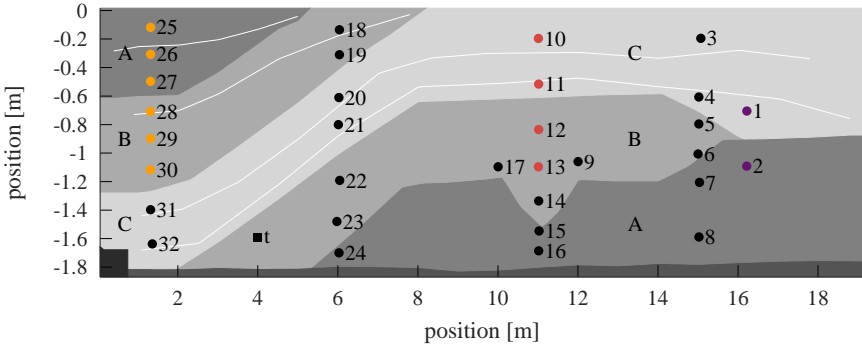

**Figure 1.** ASSESS features an effective 2D architecture with three different kinds of sand (A, B, and C). The hydraulic state can be manipulated with a groundwater well (at $18.2\,\mathrm{m}$) and is automatically monitored with 32 TDR sensors (dots) and one tensiometer (square, at $4.0\,\mathrm{m}$). The color of the dots associates TDR sensors with different cases of the 1D study discussed in Sect. 5.1. The gravel layer at the bottom of the site ensures a rapid water pressure distribution over the site. An L-element (black polygon at $0.4\,\mathrm{m}$) and compaction interfaces (white lines) were introduced during the construction. Additionally to those shown, GPR evidence indicates additional compaction interfaces. Note the different scales on the horizontal and the vertical axis.

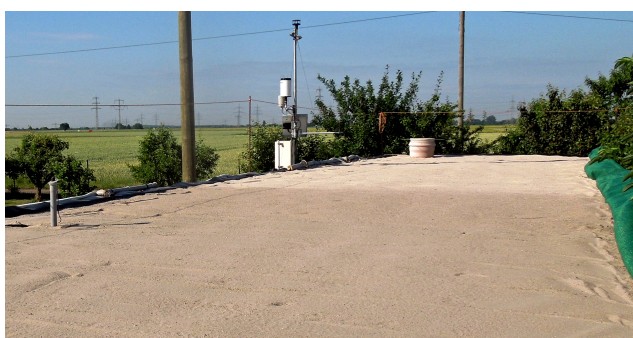

**Figure 2.** The view over ASSESS from $0 - 19\,\mathrm{m}$ shows the tensiometer, the weatherstation, and the groundwater well (left to right) as well as the color of the different sand types (figure adapted from Jaumann (2012)).

of the appropriate parameterization type for the hydraulic material properties (Dagenbach et al., 2013), and high precision monitoring of fluctuating groundwater table and infiltration experiments (Klenk et al., 2015a, b).

## 3   Representation of ASSESS

For representing the soil water dynamics in ASSESS during the experiment, we follow the lines presented by Bauser et al. (2016) and define the *representation of a system* as a set consisting of: dynamics (mathematical description), subscale physics (material properties), forcing (superscale physics) and states. We close this section by discussing the implementation of the representation.





### 3.1 Dynamics

The Richards equation (Richards, 1931),

$$\partial_t \theta_w - \nabla \cdot [K_w(\theta_w)[\nabla h_m(\theta_w) - \boldsymbol{e}_z]] = 0, \tag{1}$$

is the standard model to describe the propagation of the volumetric water content $\theta_w$ $[-]$ in time $t$ [s] with respect to the matric head $h_m$ [m]. The solution of this partial differential equation requires the specification of material properties, namely the soil water characteristic $\theta_w(h_m)$ and the hydraulic conductivity function $K_w(\theta_w)$, which are (i) highly non-linear, (ii) varying over many orders of magnitude, (iii) showing hysteretic behavior, (iv) impossible to determine a priori, and (v) very expensive to measure directly.

The unit vector in $z$-direction $\boldsymbol{e}_z$ indicates the direction of gravity, typically pointing downwards.

### 3.2 Subscale physics

Many heuristic parameterization models exist for the soil hydraulic material properties. We choose the Mualem-Brooks-Corey parameterization (Brooks and Corey, 1966; Mualem, 1976), since it describes the materials in ASSESS appropriately (Dagenbach et al., 2013).

Brooks and Corey (1966) parameterized the soil water characteristic $\theta_w(h_m)$ with a saturated water content $\theta_{w,s}$ $[-]$, a residual water content $\theta_{w,r}$ $[-]$, a scaling parameter $h_0$ $[m]$ related to the air entry pressure ($h_0 < 0$ m) and a shape parameter $\lambda$ $[-]$ related to the pore size distribution ($\lambda > 0$). Neglecting hysteresis, this parameterization may be inverted for $\theta_{w,r} \leq \theta_w \leq \theta_{w,s}$, leading to

$$h_m(\theta_w) = h_0 \left( \frac{\theta_w - \theta_{w,r}}{\theta_{w,s} - \theta_{w,r}} \right)^{-1/\lambda}. \tag{2}$$

Inserting the Brooks-Corey parameterization into the hydraulic conductivity model of Mualem (1976), yields the Mualem-Brooks-Corey parameterization for the hydraulic conductivity function

$$K_w(\theta_w) = K_{w,0} \left( \frac{\theta_w - \theta_{w,r}}{\theta_{w,s} - \theta_{w,r}} \right)^{\tau + 2 + 2/\lambda}, \tag{3}$$

which includes the saturated hydraulic conductivity $K_{w,0}$ $[\text{m s}^{-1}]$ and a fudge factor $\tau$ $[-]$ in addition to the parameters $\theta_{w,r}$, $\theta_{w,s}$, and $\lambda$.

Small-scale heterogeneities, i.e. the texture of the porous medium, can be represented with Miller scaling, if the pore spaces at any two points are assumed geometrically similar (Miller and Miller, 1956). Scaling the macroscopic reference state $h_m^*(\theta_w)$, $K_w^*(\theta_w)$ with a local characteristic length $\xi$ $[-]$, leads to locally scaled material functions (Roth, 1995):

$$h_m(\theta_w) = h_m^*(\theta_w) \cdot \xi, \qquad K_w(\theta_w) = K_w^*(\theta_w)/\xi^2. \tag{4}$$

### 3.3 Forcing

The experiment presented in this work investigates the evolving hydraulic state which is forced with a fluctuating groundwater table. The boundary condition is separated into three phases: (i) initial drainage phase, (ii) multistep imbibition phase,





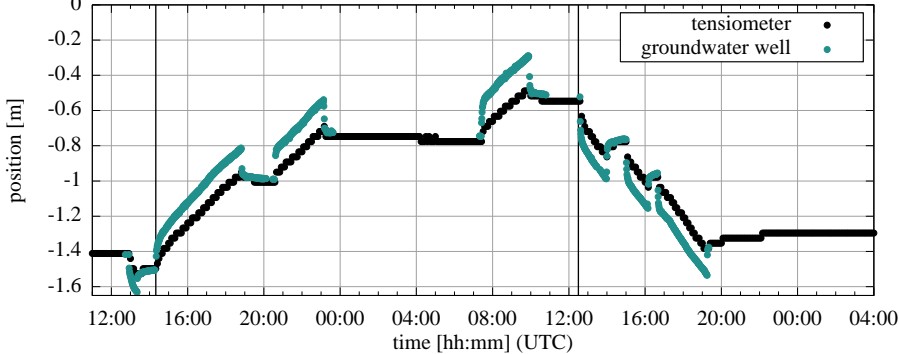

**Figure 3.** During the experiment with three different phases (initial drainage, multistep imbibition, and multistep drainage – separated by the vertical black lines in the figure), the position of the groundwater table was measured manually in the groundwater well (at $18.2$ m) and automatically with the tensiometer (at $4.0$ m). The pressure gradient between the groundwater well and the test site, i.e. the tensiometer, drives the water flux. The largest part of this pressure gradient equilibrates approximately within 5 minutes. Afterwards, the position of the groundwater table still changes, which is due to the long-term equilibration of water content distribution. Note that the discrete measurement steps reflect the resolution of the tensiometer.

and (iii) multistep drainage phase (Fig. 3). The position of the fluctuating groundwater table is measured manually[3] in the groundwater well (at $18.2$ m) and with the tensiometer (at $4.0$ m). During the multistep imbibition phase, $17.8$ m$^3$ water were pumped into the groundwater well in $9.6$ h. The equilibration steps in between were included, such that the relaxation of the capillary fringe happened within the measurement range of the TDR sensors. During the multistep drainage phase, $13.9$ m$^3$ were pumped out of the groundwater well in $5.2$ h. The detailed setup of the forcing is presented in Table 2.

### 3.4 State

The experiment was monitored in particular with soil temperature, hydraulic potential, TDR, and GPR measurements. In this work, we focus on TDR and hydraulic potential measurement data.

The evaluated relative permittivity $\varepsilon_{\mathrm{b}}$ is converted to water content $\theta_{\mathrm{w}}$ with the Complex Refractive Index Model (CRIM) (Birchak et al., 1974):

$$\varepsilon_{\mathrm{b}}(\theta_{\mathrm{w}}, T, \phi)^{\alpha} = \theta_{\mathrm{w}} \cdot \varepsilon_{\mathrm{w}}(T)^{\alpha} + (\phi - \theta_{\mathrm{w}}) \cdot \varepsilon_{\mathrm{a}}^{\alpha} + (1 - \phi) \cdot \varepsilon_{\mathrm{s}}^{\alpha}. \tag{5}$$

According to Roth et al. (1990), we set the geometry parameter $\alpha$ to $0.5$. In order to apply the CRIM, the porosity $\phi$, the relative permittivity of water $\varepsilon_{\mathrm{w}}$, the relative permittivity of air $\varepsilon_{\mathrm{a}}$, and the relative permittivity of the soil matrix $\varepsilon_{\mathrm{s}}$ have to be known. The relative permittivity of air $\varepsilon_{\mathrm{a}}$ was set to $1.0$. Assuming that the sand matrix consists mainly of Quartz (SiO$_2$) grains, the relative permittivity of the soil matrix $\varepsilon_{\mathrm{s}}$ was set to $5.0$ (Carmichael, 1989). Corresponding to core-cutter measurements, the porosity of the materials A, B, and C was assumed as $0.41$, $0.36$, and $0.38$, respectively. Following Kaatze (1989), we

---

[3]The position of the groundwater table was measured with a measurement band at the rim of the groundwater well.





parameterize the dependency of the relative permittivity of water $\varepsilon_{\mathrm{w}}$ on the soil temperature $T$ [$^\circ$C] with

$$\varepsilon_{\mathrm{w}}(T) = 10.0^{1.94404 - T \cdot 1.991 \cdot 10^{-3}}. \tag{6}$$

The measured water content data of those sensors, which were desaturated during the experiment, are displayed in Fig. 4. Due

to the small measurement volume (Robinson et al., 2003) and the narrow transition zone during imbibition (Dagenbach et al., 2013; Klenk et al., 2015a), the measured water content increases fast as the groundwater table reaches the TDR sensor. It is worth noting that if the material is not saturated at the position of the TDR sensor, the measured water content either decreases or increases during the equilibration phases, depending on the hydraulic state at this position with respect to the hydraulic equilibrium.

We attribute the spread of the water content during saturation mainly to small-scale heterogeneities and *quasi saturation* because of entrapped air (Christiansen, 1944). In order to avoid effects related to entrapped air and also two-phase flow, all TDR measurement data with an air content below 0.1 (Faybishenko, 1995) are neglected subsequently.

### 3.5 Implementation

The implementation is an integral part of the representation of the hydraulic system. In order to separate the more general

theoretical considerations from the application dependent details of the implementation, these are not directly given in each of the subsections above, but are gathered in this separate section.

#### 3.5.1 Richards equation solver

The numerical solution of the Richards equation is based on $\mu\varphi$ (muPhi, Ippisch et al., 2006) which uses a cell centered finite volume scheme with full upwinding in space and an implicit Euler scheme in time. The nonlinear equations are linearized with

an inexact Newton-Method with line search and the linear equations are solved with an algebraic multigrid solver.

#### 3.5.2 Orientation of ASSESS

It is important to consider that ASSESS is not built completely rectangular. Most importantly, both the surface and the ground are not horizontal but primarily inclined towards the groundwater well. This leads to a deviation of $\approx 0.1$ m over the length of the site. Since the applied Richards solver $\mu\varphi$ demands a rectangular structured grid, the geometry was rotated. This rotation was compensated by a counter-rotation of the gravity vector $\boldsymbol{g} \approx (0.0708, -9.8097)^\top$.

#### 3.5.3 Boundary condition

Generally, the boundary of the simulation is implemented with a Neumann no-flow condition. However, during the forcing phases, we prescribe the measured groundwater table as Dirichlet boundary condition at the position of the groundwater well. Due to measurement uncertainties, possible mixture of neighboring materials, and subsidence after the construction, Antz (2010) and Buchner et al. (2012) assess the uncertainty concerning positions of sensors and material interfaces in ASSESS to





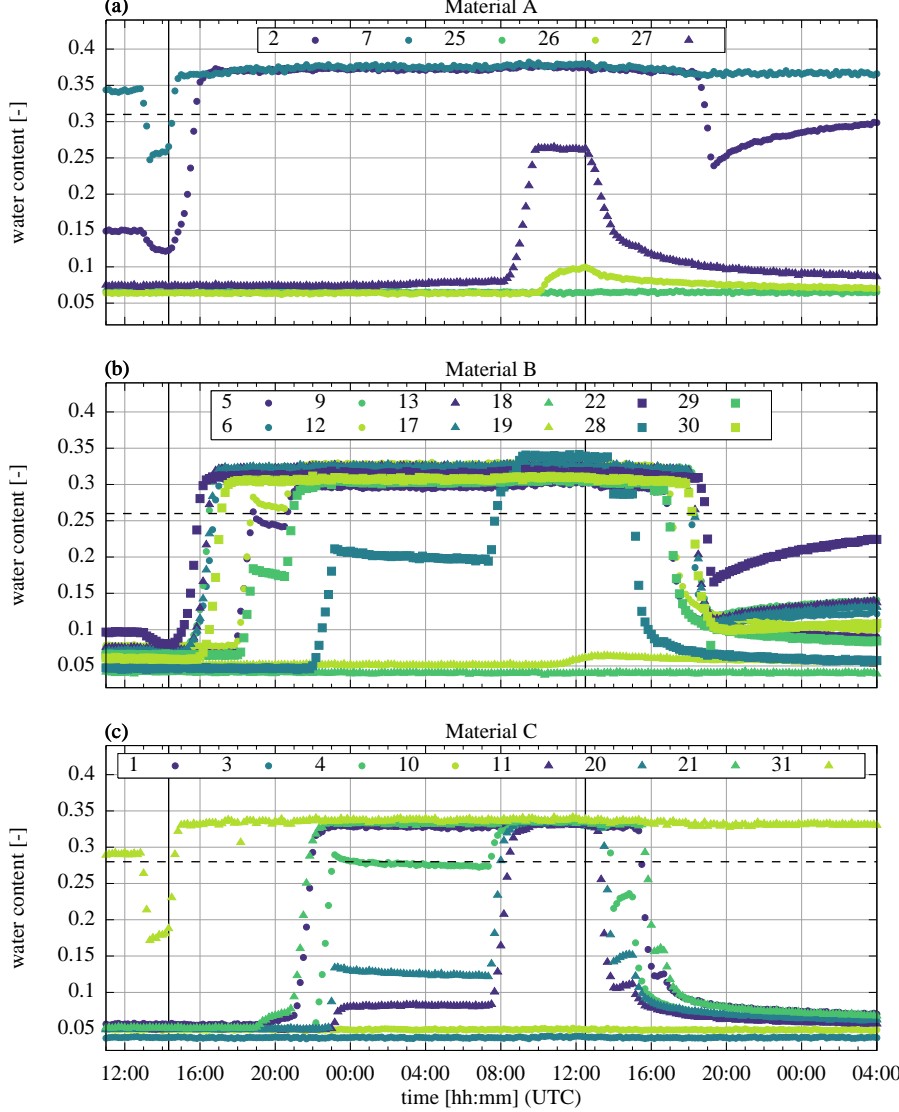

**Figure 4.** Due to the fluctuating groundwater table (Fig. 3), the water content measurement data for the three different phases (initial drainage, multistep imbibition, and multistep drainage – separated by the vertical black lines in the figure) show a high variability up to and beyond the validity limits of the Richards equation. In order to avoid effects related to entrapped air and two-phase flow phenomena, we neglect all data with a volumetric air content smaller than 0.1 (all values above the dashed lines).

$\pm\,0.05$ m. Additionally to the orientation of ASSESS (Sect. 3.5.2), this uncertainty directly translates to an uncertainty in the
5   Dirichlet boundary condition. Since input errors have a large impact on the resulting parameters, we implemented an optional
offset to the Dirichlet boundary condition which can be estimated.





### 3.5.4 Initial state estimation

Since we will use inversion methods for parameter estimation, starting the as near as possible to the measured initial state is key. Usually, this is achieved with a spin-up phase. However, for some of our investigations, a spin-up phase would exceed the computational resources available to us[4]. Hence, we developed a method to estimate the initial water content distribution based on TDR measurement data.

In the first step, we assume static hydraulic equilibrium and approximate the matric potential at the position of the TDR sensors with the negative distance of the sensor to the groundwater table. Subsequently, the approximated matric potential is associated with the measured water content for each sensor. Further, we assume spatially homogeneous and temporally constant material properties which allows us to group the TDR sensors – together with the approximated matric potential and the measured water content – by material. For each material, we then fit the parameters $h_0$, $\lambda$, and $\theta_{w,r}$ of the Brooks-Corey parameterization[5] to the approximated matric potential and the measured water content (Fig. 5). This yields an approximation for the initial water content distribution between the TDR sensors. With the resulting parameter values for each material, the subsurface material distribution, and the position of the groundwater table, we can calculate an estimation of the initial water content distribution in ASSESS (Fig. 6).

As the parameters for the Brooks-Corey parameterization are derived from measurement data, we may also use them as initial parameter values for computationally expensive gradient-based inversions (Sect. 5.2). The missing initial values for the parameters $\tau$ and $K_{w,0}$ are taken from Carsel and Parrish (1988) in this work[6]. We will refer to these parameters as *initial state material functions* in the remainder of this work.

In particular due to (i) a limited number of TDR sensors, (ii) missing hydraulic potential measurements at the position of the TDR sensors, and (iii) spatial small-scale heterogeneity present in the materials, structural differences between the estimation and the measurements occur which indicate limitations of describing ASSESS with effective soil hydraulic material properties[7].

### 3.5.5 Small-scale heterogeneity and TDR measurement volume

In order to represent the small-scale heterogeneity of the material properties, the center of each grid cell is associated with a Miller scaling factor that is initialized to $1.0$. As the information about the small-scale heterogeneity only enters via the TDR measurement data, the exact position of each TDR sensor is also associated with a Miller scaling factor. For each TDR sensor,

---

[4]Depending on the hydraulic material properties, the 45 h forward simulation of the 2D study presented in Sect. 5.2 took 0.25–1.0 h at low grid resolution. The parameter estimation for this case took about 3–4 days on a cluster with as many cores as parameters. A proper spin-up phase would at least cover a month, increasing the simulated time to $45\,\mathrm{h} + 30 \cdot 24\,\mathrm{h} = 765\,\mathrm{h}$. This would increase the computation time up to a factor of 17, namely 4.25–17.0 h per forward run and approximately 51–68 days for the inversion on a cluster with as many cores as parameters.

[5]The saturated water content $\theta_{w,s}$ is assumed to be known from core-cutter measurement data.

[6]We used the parameter set *sand* with $\tau = 0.5$ and $K_{w,0} = 8.3 \cdot 10^{-5}\,\mathrm{m\,s}^{-1}$.

[7]Additional insight can be gained by closely investigating the structural deviation of the measured water content of TDR sensors 5, 12, and 29 from the estimation of the initial state for material B in Fig. 5. Klenk et al. (2015a, Fig. 1b and 6) presented GPR measurements, which indicate that at least TDR sensors 6, 9, 13, 17, and 22 are closely below a compaction interface and thus are experiencing a compacted pore structure. This can explain, why these TDR sensors measure smaller water content values compared to the ones measured by the TDR sensors 5, 12, and 29.





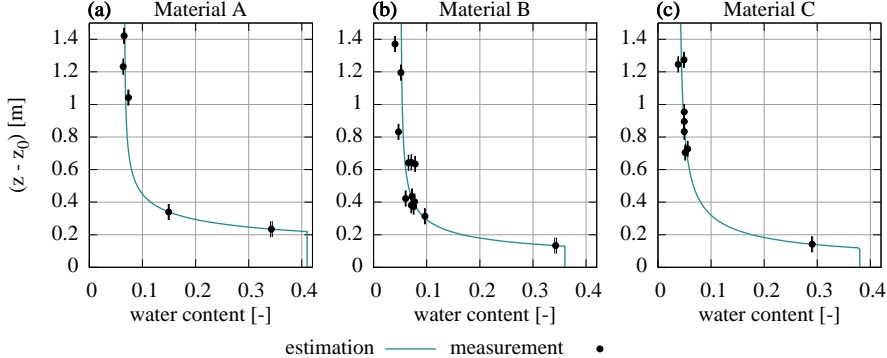

**Figure 5.** We use the Brooks-Corey parameterization to estimate the initial water content distribution between the TDR sensors. Assuming hydraulic equilibrium, we approximate the matric potential $h_\mathrm{m}$ with the negative distance to the groundwater table position $z_0$: $h_\mathrm{m} \approx -(z - z_0)$. For each material, we then use the approximated matric potential at the position of the TDR sensors and the corresponding water content measurement data to fit the Brooks-Corey parameters. Each dot depicts the mean of 15 subsequent data points measured in the 4 h preceding the experiment. The according standard deviations are all smaller than 0.0017, which indicates (i) that the hydraulic system is relatively equilibrated at the beginning of the experiment and (ii) that the deviations from the estimation are statistically significant.

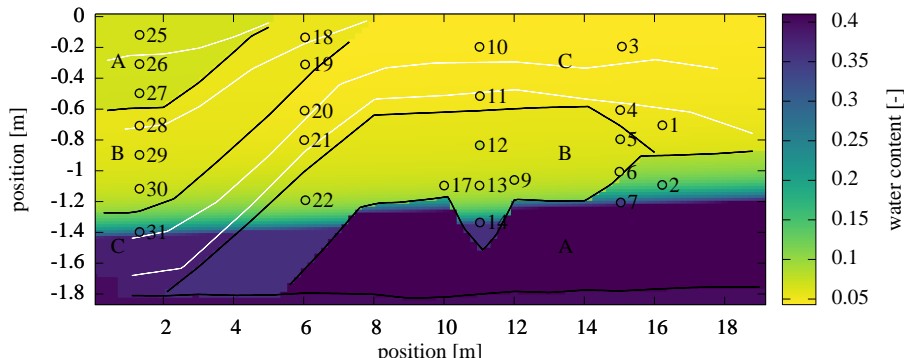

**Figure 6.** The estimated initial water content distribution is based on the TDR measurement data (face color of the circled dots, Fig. 5). Since the saturated water content $\theta_\mathrm{w,s}$ is fixed for each material a priori, only TDR sensors in unsaturated material are shown. Due to the orientation of ASSESS (Sect. 3.5.2), the groundwater table is slightly slanted. The black lines indicate material interfaces, whereas the white lines indicate compaction interfaces, which were introduced during the construction of ASSESS. Additionally to those shown, GPR evidence indicates additional compaction interfaces. Note the different scales on the horizontal and the vertical axis.

we implemented a bivariate Gaussian distribution, which determines the scaling factors in the neighborhood of this sensor. The distribution is centered at the position of the sensor, has a standard deviation of $0.015\,\mathrm{m}$ in horizontal as well as in the vertical direction, and approaches $1.0$ with increasing distance from the TDR sensor. Finally, these distributions determine the Miller scaling factor at the center of each grid cell.





Since imbibition fronts can be very steep, we implemented the measurement volume of the TDR sensors by averaging the simulation data within a measurement radius of $0.015$ m.

## 4  Parameter estimation

For estimating parameters, we employ the Levenberg-Marquardt algorithm. We include this locally convergent algorithm in a local-global approach, in order to analyze the convergence behavior or if no suitable initial parameters are available. Therefore, we generate an ensemble of the initial parameter sets with a Latin Hypercube algorithm[8]. As the sampled initial parameter sets are uniformly distributed in parameter space, the convergence path and the resulting parameter sets of the Levenberg-Marquardt algorithm contain much information regarding the convergence radius and the distribution of local minima.

After discussing the Levenberg-Marquardt algorithm and the general setup of the inversions, we close the section with a structural error analysis.

### 4.1  Levenberg-Marquardt algorithm

Our implementation of the Levenberg-Marquardt algorithm[9] is mainly based on Moré (1978), Press (2007), and Transtrum and Sethna (2012). As it additionally includes some modifications, it is sketched shortly in the following.

Assuming $M$ data points $m_\mu$ $(1, 2, \ldots, \mu, \ldots, M)$ measured at position $\boldsymbol{x}_\mu$ with standard deviation $\sigma_\mu$ and a model $f$ with $P$ parameters $p_\pi$ $(1, 2, \ldots, \pi, \ldots, P)$, the $\chi^2$ cost function is defined as

$$\chi^2(\boldsymbol{p}) = \frac{1}{2} \sum_{\mu=1}^{M} \left( \frac{m_\mu - f(\boldsymbol{x}_\mu, \boldsymbol{p})}{\sigma_\mu} \right)^2 = \frac{1}{2} \sum_{\mu=1}^{M} r_\mu(\boldsymbol{p})^2. \tag{7}$$

It implicitly assumes statistically independent random representation errors which are normally distributed with zero mean. The standardized residuals $r_\mu$ can be expanded

$$r_\mu(\boldsymbol{p} + \delta\boldsymbol{p}) \approx r_\mu(\boldsymbol{p}) + \sum_{\pi=1}^{P} J_{\mu\pi} \delta p_\pi \tag{8}$$

with the Jacobi matrix $J_{\mu\pi} = \partial r_\mu / \partial p_\pi$. The Jacobi matrix is assembled numerically with the finite differences method which allows for trivial parallelization of the required $P$ forward runs. Following Press (2007), the Hessian is approximated ($\mathbf{H} \approx \mathbf{J}^\top \mathbf{J}$), assuming that the second term in the derivative cancels out as $f(\boldsymbol{x}_\mu, \boldsymbol{p}) \to m_\mu$ with increasing number of iterations. For the Gauss-Newton algorithm then follows

$$\delta\boldsymbol{p} = -(\mathbf{J}^\top \mathbf{J})^{-1} \cdot \nabla \chi^2(\boldsymbol{p}). \tag{9}$$

Since $\mathbf{J}^\top \mathbf{J}$ does not always have full rank, the inversion may be ill conditioned leading to uncontrolled large steps. One possibility to cope with this issue, is to regularize $\mathbf{J}^\top \mathbf{J}$ by adding a diagonal damping matrix $\mathbf{D}^\top \mathbf{D}$. We follow Transtrum

---

[8]The sampling algorithm was implemented with the help of the pyDOE package: https://github.com/tisimst/pyDOE.

[9]Our implementation of the Levenberg-Marquardt algorithm is written in C++ and employs the Eigen library (Guennebaud et al., 2010).





and Sethna (2012) and choose this damping matrix, such that the diagonal entry for $p_\pi$ contains the corresponding maximal diagonal entry of $\mathbf{J}^\top \mathbf{J}$ from all previous iterations if this value is larger than a predefined minimal value (1.0) which is used otherwise. The resulting damping matrix is scaled with a parameter $\lambda$ which tunes both the amount of regularization and the step size of the parameter update.

Finally, the parameter update $\delta \boldsymbol{p}$ is calculated via

$$\delta \boldsymbol{p} = -(\mathbf{J}^\top \mathbf{J} + \lambda \cdot \mathbf{D}^\top \mathbf{D})^{-1} \cdot \nabla \chi^2(\boldsymbol{p}), \tag{10}$$

where the linear problem is solved with a Singular Value Decomposition (SVD). If the condition number of the sensitivity matrix $S = \mathbf{J}^\top \mathbf{J} + \lambda \cdot \mathbf{D}^\top \mathbf{D}$ is larger than a threshold ($10^{12}$), the linear problem is solved approximately with the Conjugate Gradient algorithm by choosing the maximal number of iterations smaller than the number of parameters $P$.

The proposed parameters at iteration $i$ are finally given as

$$\boldsymbol{p}^{i+1} = \boldsymbol{p}^i + \delta \boldsymbol{p}^i. \tag{11}$$

The convergence path of the Levenberg-Marquardt algorithm is influenced by both the size of the scaling parameter $\lambda_{\text{initial}}$ and the choice how to adapt $\lambda$ after each iteration. For this work, we chose $\lambda_{\text{initial}} = 5.0$ and applied the *delayed gratification* strategy proposed by Transtrum and Sethna (2012). According to this strategy, $\lambda$ is decreased by a previously chosen factor (2.0) if the parameter update is successful and increased by a larger factor (3.0) if the update is not successful.

The described gradient-based algorithm heuristically balances performance and stability. Expanding the stability measures, we add an optional damping factor which decreases the correction of certain parameters. This damping factor is intended for parameters representing higher order uncertainties. We use this approach to estimate sensor positions and Miller scaling factors along with effective soil hydraulic properties. Therefore, we initialize sensor positions and Miller scaling factors to neutral values and set the damping factor for these parameters to 0.1. This reduces the applied correction of these parameters to 10% of the proposed correction by the Levenberg-Marquardt algorithm. Hence, the main focus of the algorithm is to estimate effective soil hydraulic properties, whereas higher order uncertainties are adjusted more gradually.

## 4.2 General setup and estimated parameters

The general setup of the parameter estimation for ASSESS (Sect. 2) is explained with Fig. 7. For each of the three materials, we estimate the Mualem-Brooks-Corey parameters $h_0$, $\lambda$, $K_{\text{w},0}$, $\tau$, and $\theta_{\text{w,r}}$ (Sect. 3.2). The saturated water content $\theta_{\text{w,s}}$ is assumed to be equal to an estimate for the porosity $\phi$ based on core-cutter measurements (Sect. 3.4). In order to avoid parameter bias due to input errors, we estimate (i) a constant offset to the Dirichlet boundary condition (Sect. 3.5.3) and (ii) the saturated hydraulic conductivity of the gravel layer. Depending on the setup (Sect. 5), we also estimate TDR and tensiometer sensor positions as well as Miller scaling factors at the position of the TDR sensors (3.5.5).





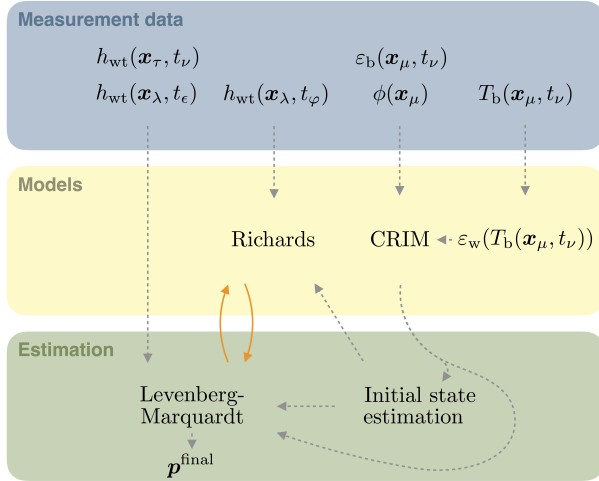

**Figure 7.** The available hydraulic potential $h_{\mathrm{wt}}$ is measured at the position of the groundwater well $\boldsymbol{x}_\lambda$ and at the position of the tensiometer $\boldsymbol{x}_\tau$. The data set, which is measured in the groundwater well, is split according to the measurement times: The data measured during the forcing phases $t_\varphi$ enter the Levenberg-Marquardt algorithm (Sect. 4.1) directly, whereas the data measured during the equilibration phases $t_\epsilon$ are only used as boundary condition for the Richards equation (Sect. 3.1). The bulk relative permittivity $\varepsilon_{\mathrm{b}}(\boldsymbol{x}_\mu, t_\nu)$ and the bulk soil temperature $T_{\mathrm{b}}(\boldsymbol{x}_\mu, t_\nu)$ are measured at the position of the TDR sensors $\boldsymbol{x}_\mu$ at times $t_\nu$. Together with the porosity $\phi(\boldsymbol{x}_\mu)$, these data are transferred to water content data (Sect. 3.4), which enter the initial state estimation (Sect. 3.5.4) yielding an initial water content distribution and optional initial parameter values for the Levenberg-Marquardt algorithm. Additionally, the water content data are also directly used in the Levenberg-Marquardt algorithm. Dashed grey arrows represent one-time preparation steps, whereas solid orange arrows represent the iterative steps of the Levenberg-Marquardt algorithm yielding the final material parameters $\boldsymbol{p}^{\mathrm{final}}$.

## 4.3 Structural error analysis

Quantitative learning about complicated systems is an iterative process (Box et al., 2015; Gupta et al., 2008). Starting from conceptual ideas, the modeler represents the current understanding of the system with a model incorporating decisions and underlying hypotheses (Clark et al., 2011; Gupta et al., 2012). The optimal experimental design addressing specific research objectives is based on the model and thereby on the current understanding of the system. The resulting measurement data reveal the answer of reality to specific questions posed by the experimentator. By comparing the forecast of the model with the measurement data, it can be investigated, how well the questioned behavior of the system is understood quantitatively. Thus, disagreement between the model and the measurement data reveals incorrect understanding of the system. Consequently, the concepts, decisions, and hypotheses with respect to the model (including measurement data evaluation procedures) and the measurement data themselves have to be revised. This leads to an improved model as well as improved measurement data acquisition and evaluation procedures. If the model predicts the measurement data accurately and precisely enough, the research objectives have to be expanded, such that the measurement data cover a larger part of the state space. This step is necessary, because high model complexity admittedly yields an accurate description of the measurement data, which, however, is forcedly





based on biased and case dependent parameters. Ultimately, this iterative procedure leads to measurement data covering the whole state space and a statistical model-data mismatch corresponding to the measurement data error model – indicating complete understanding of reality. In general, however, such measurement data are not available and the application merely requires a limited accuracy and precision. Hence, determining the sufficient complexity of the model and the measurement data for the required accuracy and precision is the crux.

By applying the $\chi^2$ cost function (Eq. (7)), it is implicitly assumed that the model is perfect aside from a white Gaussian noise. This corresponds to complete understanding of reality and a Gaussian measurement data error model. Structural model-data mismatch indicates that this assumption is invalid. One way to quantify this problem is to analyze the total uncertainty space with a Bayesian total error analysis (BATEA) (Kavetski et al., 2002, 2006). In our case, a Bayesian analysis of the total uncertainty space is not feasible, primarily due to a lack of models, e.g., for hysteresis. Instead, we neglect highly complicated

representation errors in the hope that if their representation is necessary, the structural model-data mismatch will reveal this. The contribution of representation errors, which could not be quantified or excluded from the measurement data a priori, is parameterized and explicitly estimated. Table 3 gives an overview over the treatment of the considered representation errors. Structural deviations from the measurement data or prior estimates for the parameters, which remain after the optimization, hint at representation errors which should be corrected in the subsequent iteration.

The structural error analysis and the assessment of uncertainties results from iterative preliminary evaluations. In order to showcase the power of the method and the sensitivity of the fluctuating groundwater table experiment, we shortly present the results of one of those preliminary evaluations. In this case, the orientation of ASSESS (Sect. 3.5.2) was not yet compensated for by rotating the geometry and the gravitation vector. Considering the structural error analysis, we parameterized and estimated uncertain contributions to the representation. Hence, not only the Mualem-Brooks-Corey parameters, an offset to the

Dirichlet boundary condition and the saturated hydraulic conductivity of the gravel layer, but also the position of the TDR sensors were estimated. The results presented in Fig. 8 show that the estimated TDR positions display a consistent deviation from the measured positions as they compensate for the orientation of ASSESS. Thus, the position of most sensors on the right is estimated to be higher and the position of most sensors on the left is estimated to be lower than the measured ones. By estimating the sensor position, we also incorporated other representation errors into the resulting parameters, such as small-

scale heterogeneities and eventually a non-represented evaporation front mostly affecting the estimated position of the upper sensors (3, 11, 18, and 25). Hence, this analysis (i) demonstrates the difficulty to separate representation errors and (ii) is able to identify representation errors which have to be improved subsequently.

Being key for the identification of representation errors, a visual analysis of the standardized residual increases the intuitive understanding of the model-data mismatch (e.g., Legates and McCabe, 1999; Ritter and Muñoz-Carpena, 2013). Therefore, the standardized residual is visualized over time and over the theoretical quantiles corresponding to a Gaussian distribution with the standard deviation of the measurement data. The former visualization highlights the structural model-data mismatch and the latter permits easy comparison of the standardized residual distribution to the expected Gaussian distribution. Additionally,

5  statistical measures help to benchmark the model-data mismatch. As single measures might be misleading (Legates and Mc-





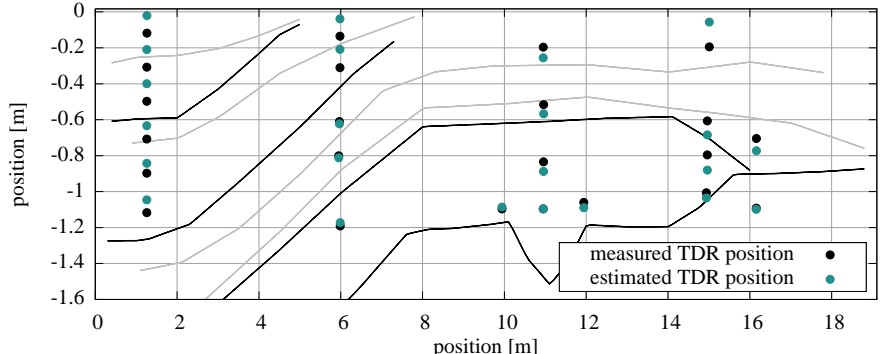

**Figure 8.** Comparison of measured and estimated TDR sensor positions based on an exemplary preliminary evaluation of the measurement data. The consistent deviation of the estimated TDR sensor positions reveal an unrepresented model error: The orientation of ASSESS (Sect. 3.5.2). The black lines indicate material interfaces, whereas the grey lines indicate compaction interfaces, which were introduced during the construction of ASSESS. Additionally to those shown, GPR evidence indicates additional compaction interfaces. Note the different scales on the horizontal and the vertical axis.

Cabe, 1999), we apply (i) the root mean square error ($e_{\mathrm{RMS}}$), (ii) the mean absolute error ($e_{\mathrm{MA}}$), and (iii) the Nash–Sutcliffe model efficiency coefficient ($e_{\mathrm{NS}}$) (Nash and Sutcliffe, 1970).

## 5 Case studies

In this section, we analyze the estimation of effective material properties for ASSESS based on a 1D and a 2D study. For each of these studies, four different setups were implemented: (i) *naive*: We estimate the hydraulic material properties, an offset to the Dirichlet boundary condition, and the saturated hydraulic conductivity of the gravel layer. (ii) *position*: In addition to the parameters estimated in the *naive* setup, we also estimate the sensor positions. (iii) *miller*: In addition to the parameters estimated in the *naive* setup, we estimate one Miller scaling factor for each TDR sensor. (iv) *miller and position*: In addition to the parameters estimated in the *naive* setup, we estimate both the sensor positions and one Miller scaling factor for each TDR sensor.

### 5.1 1D study

In order to investigate whether the experiment at ASSESS can be described with a 1D model, we set up three different cases with an increasing number of TDR sensors per material (Table 4): The *case I* includes the measurement data of sensor 1 in material C and sensor 2 in material A, and thus comprises one sensor per material. The *case II* includes two sensors per material, namely the sensors 10 and 11 in material C and sensors 12 and 13 in material B. Finally, the *case III* includes three sensors per material, namely the sensors 25, 26, 27 in material A and sensors 28, 29, 30 in material B. Note that the cases are located





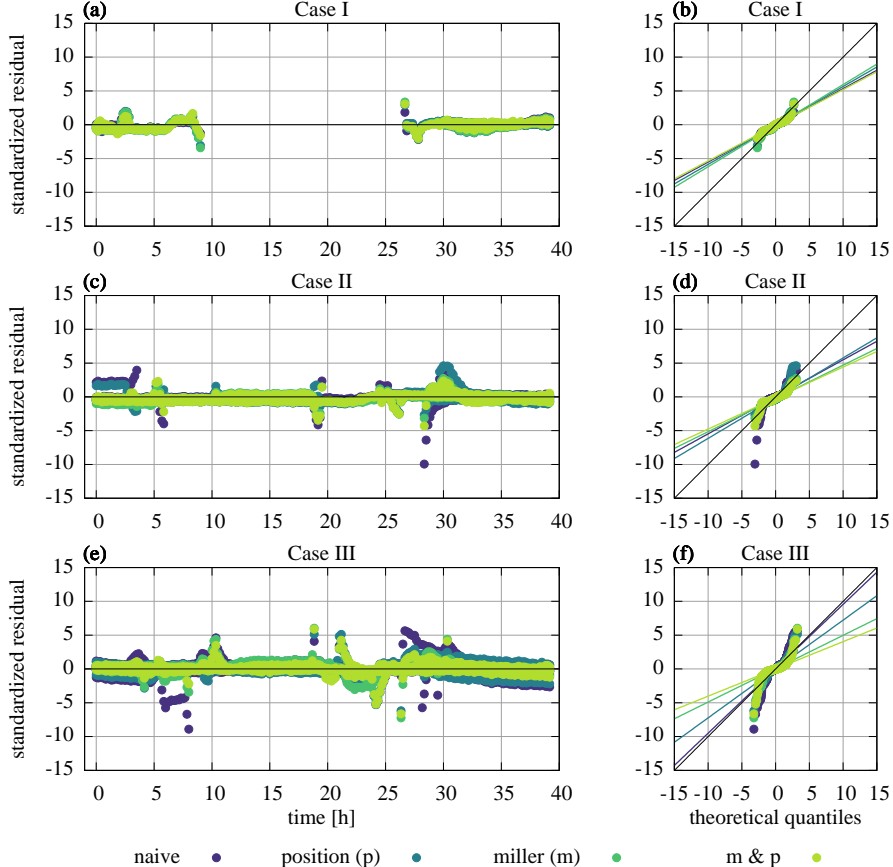

naive   •   position (p)   •   miller (m)   •   m & p   •

**Figure 9.** For the 1D study, the standardized residuals of the best ensemble member are visualized over time (left) and over the theoretical quantiles of a Gaussian with the estimated standard deviation of the TDR measurements (0.007) (right). The the cases are analyzed with four setups *naive*, *position*, *miller*, and *miller and position*. The more sensors per material are used in the inversion, the worse the representation of the *naive* setup gets. In this case, representing uncertainties with respect to the sensor position and small-scale heterogeneities improves the representation substantially. The decreasing slope of a linear fit (thin lines in the probability plots), which is based on the standardized residuals within $[-2, 2]$ theoretical quantiles, also indicates this improvement.

5    at different positions in ASSESS (Fig. 1).

As described above, the analysis is organized in four different setups (*naive*, *position*, *miller*, and *miller and position*). The *naive* setup is adjusted for the 1D studies, such that not only the material functions of the materials with sensors, but also the saturated conductivity of the third material[10] are estimated for *case II* and *case III*. The other setups remain accordingly.

We use the manually measured groundwater table data as Dirichlet boundary condition. Uncertainties concerning the position of the sensors and the subsurface material interfaces (Sect. 3.5.3) directly translate to uncertainties in the boundary condition.

---

[10]Material A in *case II* and material C in *case III*



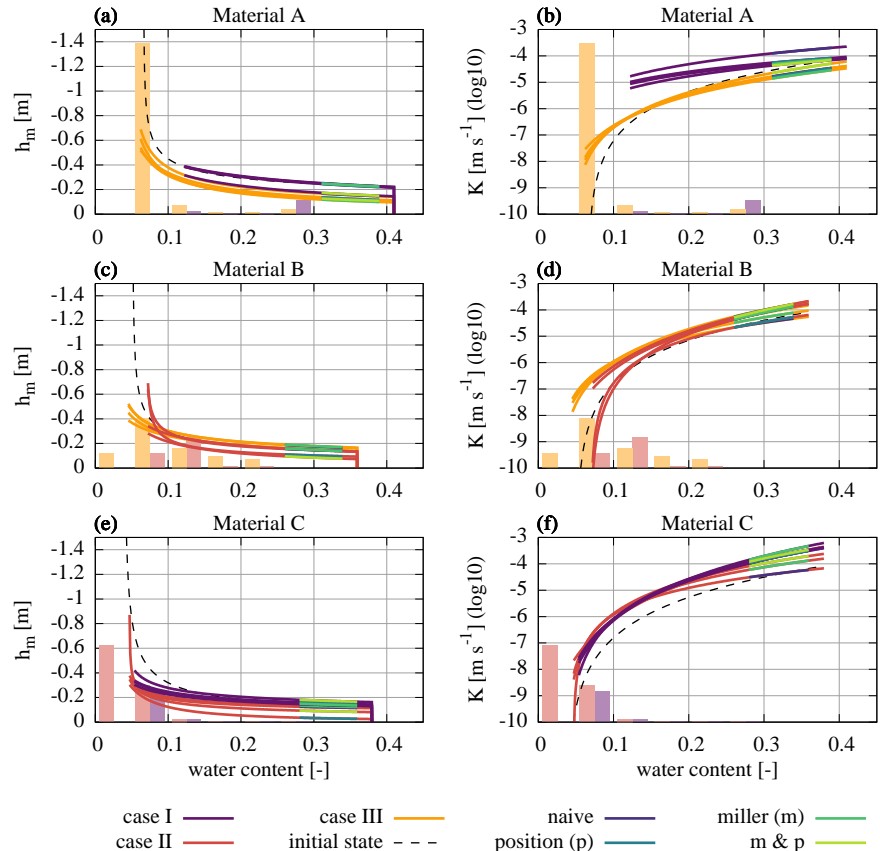

**Figure 10.** The 1D study consists of three cases (*case I*, *case II*, and *case III*). Together with the material functions resulting from the ***initial state estimation*** (Sect. 3.5.4), we visualize the resulting material functions of the best ensemble member for each setup (*naive*, *position*, *miller*, or *miller and position* – denoted by the color close to saturation). For all inversion results, the plot range is adjusted to the available water content range. The number of water content measurements within intervals of 0.05 is indicated with histogram bars for each case. The height of these bars is normalized over all figures. The main message of this figure is, that unrepresented model errors may lead to biased parameters.

Accounting for the orientation of ASSESS (Sect. 3.5.2), we add a constant offset to the Dirichlet boundary condition for each case (*case I*: -0.02 m, *case II*: -0.05 m, *case III*: -0.12 m). In order to minimize the input error, we also estimate this offset in the inversion.

If TDR sensor positions are estimated, these are initialized to the measured position. Similarly, the Miller scaling factors are initialized to 1.0. The forward simulations were calculated on a grid with $1 \times 400$ cells on 1.9 m and $10^{-8}$ as limit of the Newton solver (Sect. 3.5.1). Following Jaumann (2012), the standard deviation of the TDR measurements is assumed as 0.007. Since the hydraulic potential is not measured within the domain of these 1D studies, the inversions are only based on the





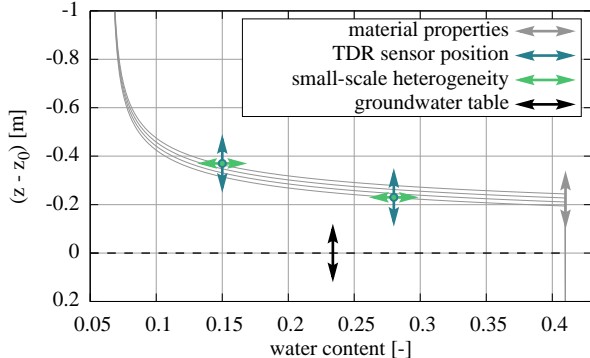

**Figure 11.** In this sketch, we visualize the uncertainties with respect to (i) the material properties, (ii) the TDR sensor position, (iii) the small-scale heterogeneity, and (iv) the groundwater table position. During static phases, these uncertainties can lead to correlated estimated parameters, e.g., as an incorrect position of the groundwater table can be compensated by changing $h_0$ *and* $\lambda$. During transient phases, however, the addressed uncertain model components have distinct effects on the model, e.g., as $\lambda$ also changes the conductivity function. Hence, the ability of the parameter estimation algorithm to separate these uncertain model components depends on the available measurement data.

TDR water content measurements. For each of the different setups, we ran an ensemble of 20 inversions starting from Latin-
5  Hypercube sampled initial parameter sets. In each setup, we only analyze the ensemble member with minimal $\chi^2$ in the subsequent evaluation. The according statistical measures ($e_{\mathrm{RMS}}$, $e_{\mathrm{MA}}$, and $e_{\mathrm{NS}}$) are given in Table 5. Here, we only refer to the $e_{\mathrm{MA}}$, because the $e_{\mathrm{RMS}}$ and the $e_{\mathrm{NS}}$ are behaving accordingly if not stated differently.

Combining the data of all applied TDR sensors, the standardized residual for each case is presented in Fig. 9.
10  Investigating the resulting standardized residuals of *case I*, it is striking that all setups describe the measurement data qualitatively equally well. Since the estimation of the material properties is only based on one sensor per material in this case, the material parameterization offers enough freedom to describe the measurement data. Hence, it also describes unrepresented model errors, such as the sensor position and small-scale heterogeneities. Therefore, additional representation and estimation of TDR sensor positions or Miller scaling factors do not lead to further improvement. The largest residuals occur during highly
15  transient phases. Compared to the measurement data, the simulated imbibition phase is too slow for sensor 1 and too fast for sensor 2. Also the simulated drainage phase is too slow for sensor 1 and drainage behavior of sensor 2 is consistently wrong. This structural model-data mismatch hints at unrepresented model errors due to the restriction to a 1D domain. Yet, the residuals of all setups are smaller than 5 standard deviations, which translates to 3.5 % volumetric water content.

We noted in section 4.3 that by applying the $\chi^2$ cost function, we implicitly assume that the model can describe the measurement data up to a white Gaussian noise. However, this is generally not the case, because the measurement error may include a bias (accuracy and precision) and the representation might neglect processes in the dynamics, for example. Inspecting the probability plots of the three cases, we spot a characteristic *S*-shape: The slope < 1 for small residuals indicates that the preci-





sion of the simulation is smaller than the standard deviation of the Gaussian distribution with the standard deviation of the TDR

measurements. The slope $> 1$ for large residuals shows that these residuals are larger than the presumed Gaussian distribution. Generally, the *S*-shape indicates non-Gaussian distributions. Since the large residuals are of structural instead of random nature and because the large residuals preferably occur in transient phases, we attribute them mainly to missing processes in the dynamics or biased parameters. As the curves are basically centered at the origin, a significant constant bias can be excluded. The $e_{\text{MA}}$ of the *naive* setup increases in *case II*, because there are two sensors per material and the effective material pa-

rameterization can not completely compensate for the small-scale heterogeneity at the position of both sensors. Consistently, representing the small-scale heterogeneity improves the description of the measurement data. As before, the largest residuals occur during the highly transient phases, especially during the drainage phase. Except for two outliers, the residuals stay smaller than 5 standard deviations here as well. Considering three sensors per material in *case III*, the $e_{\text{MA}}$ increases even further in the *naive* setup. Consequently, representing small-scale heterogeneities and uncertainties in the sensor position in the

*miller and position* setup improves the $e_{\text{MA}}$ by more than a factor of 2.

In Fig. 10, the resulting material properties of the evaluated ensemble members are visualized for the respective materials. Comparing the results of the different cases and setups, we notice a vertical shift in the soil water characteristic for each material. It seems reasonable to attribute this vertical shift to the high number of estimated uncertain model components (Sect. 4.2), because during static phases and if only few measurement sensors are available, these uncertain model components can

be correlated (Fig. 11). However, during transient phases, the distinct properties of these uncertain model components are more clearly pronounced, for example as the Brooks-Corey parameter $\lambda$ and the Miller scaling factors also influence the hydraulic conductivity. If monitored close enough with TDR sensors and hydraulic potential measurements, the parameter estimation algorithm can separate the effects better leading to a more unique solution (Sect. 5.2). In order to further analyze this vertical shift, we also ran the inversions without estimating the offset to the Dirichlet boundary condition. Besides destabilizing the

convergence of the Levenberg-Marquardt algorithm due to the increased input error, this fully transfers the uncertainty in the boundary condition to the sensor position. Hence, those setups, which estimate the sensor position, clearly outperform the others. It is worth noting that not estimating the offset to the Dirichlet boundary condition does not remove the vertical shift of the soil water characteristics. Hence, as the given measurement data are merely sensitive to the curvature of the soil water characteristic, we will mainly focus on its curvature, e.g., when comparing the inversion results with the initial state material

functions in the subsequent evaluation.

The three cases cover the three materials at different locations in ASSESS and are based on distinct measurement data with respect to both quantity and measurement data range. This is most evident for material A which is located at the bottom of ASSESS and nearly saturated in *case I* whereas it is at the top and rather dry in *case III* (colored dots in Fig. 1). Thus, also the unrepresented model errors have different effects. Subsequently, we highlight one example which is most pronounced dur-

ing the final equilibration phase. In *case III*, the water content at position of the TDR sensors 25, 26, and 27 is higher than in hydraulic equilibrium, leading to a drainage flux and a decrease in water content. However, in *case I*, the TDR sensor 2 monitors the relaxation of the capillary fringe leading to an increasing water content. Due to lateral flow, this data includes the relaxation of the whole test site. The different characteristic behavior of the measurement data during the equilibration phase





is shown in Fig. 4. In order to minimize the structural model-data mismatch during the equilibration phase, the parameter

estimation algorithm increases the hydraulic conductivity to compensate for the non-represented lateral flow with vertical flow from above the sensor. This interpretation is supported by the fact that the hydraulic conductivity of *case I* is larger than the hydraulic conductivity for both the *case III* and for the 2D study is discussed in subsequent section. Material B is in the middle of ASSESS and thus the inversions of *case II* and *case III* are based on comparable measurement data. Therefore, we expect relatively congruent resulting material functions. This expectation is confirmed by the results, except for the two setups in *case*

*II*, in which no Miller scaling factor was estimated. These setups show a deviating curvature of the soil water characteristic and of the hydraulic conductivity function. This effect is explained in more detail in the subsequent section. Regarding material C, we can identify both effects – the vertical shift and the deviating curvature of the soil water characteristic. The large uncertainty in the saturated hydraulic conductivity reflects the low sensitivity of the measurement data on this parameter due to the lack of measurements influenced by the saturated material C.

Although the initial parameter sets for the 1D inversions were Latin Hypercube sampled, the curvature of the soil water characteristic for the inversion results is reasonably close the initial state material functions which allows to use the latter to initialize gradient-based inversion methods. The estimate of the initial state material function for material C deviates strongest from the inversion result compared to the other two materials, since in material C only few sensors are available to assess the form of the capillary fringe. Naturally, the better the available number of TDR sensors is spread over the water content range, the better

the fit of the initial state parameters gets (Sect. 3.5.4). Iteratively restarting the inversion using the previous inversion results as initial state material functions is likely to improve the representation. Since $K_{w,0}$ and $\tau$ are prescribed a priori and are not estimated for the initial water content distribution, the hydraulic conductivity functions associated with the initial state show large deviations from the inversion results.

In summary, we demonstrated that the more sensors per material are used in the inversion, the larger the probability gets to observe states (and model errors) which can not be described accurately and precisely enough with the *naive* setup. Naturally, this increases the size of the structural model-data mismatch. Hence, in order to avoid biased parameters, significant model errors have to be represented. The estimation of TDR sensor positions and Miller scaling factors constitutes a major step in this direction, as this decreased the $e_{MA}$ by more than a factor of 2 in the case with most sensors per material. Due the

low number of TDR sensors monitoring transient phases and the lack of hydraulic potential measurement data, the Levenberg-Marquardt algorithm was not able to completely separate the estimated uncertain model components. This effect becomes most evident in the discussed vertical shift of the soil water characteristic. We found that the restriction to a 1D domain leads to an overestimation for the hydraulic conductivity function of material A due to unrepresented lateral flow. Finally, we observed that the initial state material functions are reasonably close to the inversion results to use them as initial values for gradient-based inversion methods.





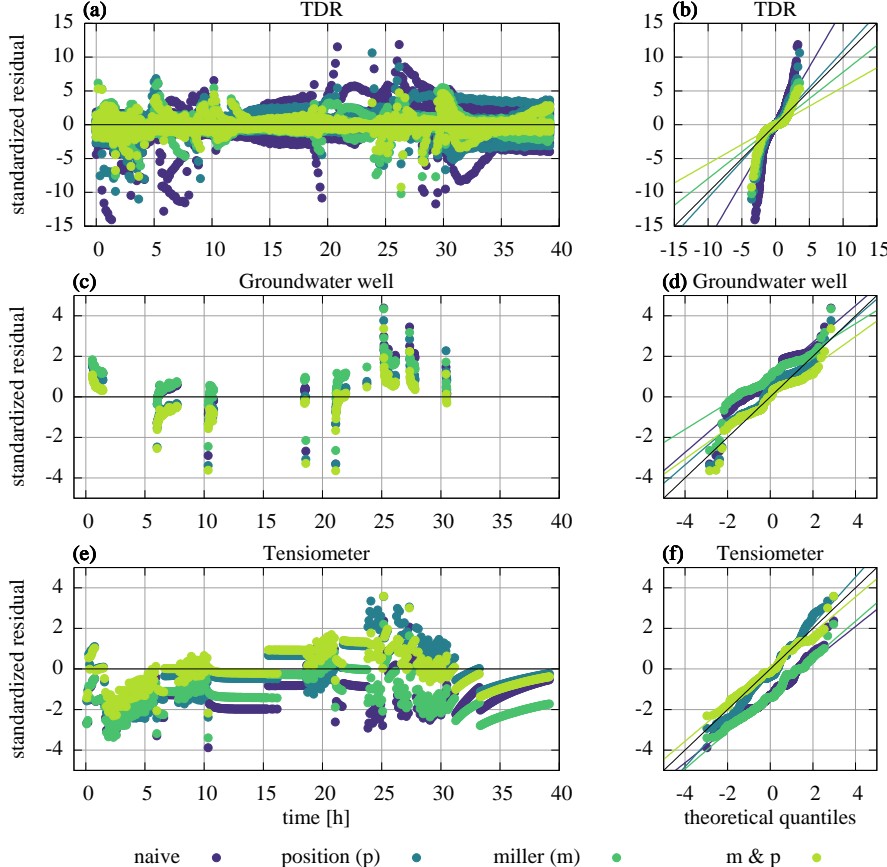

**Figure 12.** The standardized residuals of the 2D study are visualized over time (left) and in a probability plot (right) for all TDR and hydraulic potential sensors. The color associates the results with the four setups of the study (*naive*, *position*, *miller*, and *miller and position*). Same as for the 1D study, the standard deviation for the TDR measurement data is chosen as $0.007$. We choose the standard deviation for both the manual measurements in the groundwater well and the tensiometer measurement data as $0.025$ m. The representation of uncertainties with respect to the sensor positions and small-scale heterogeneities improves the description of the TDR data significantly. The decreasing slope of a linear fit (thin lines in the probability plots), which is based on the standardized residuals within $[-2, 2]$ theoretical quantiles, also indicates this improvement. The structural model-data mismatch for the hydraulic potential data is mainly due to (i) uncertainties concerning the position of the tensiometer and (ii) unrepresented 3D flow phenomena.

## 5.2 2D study

In this section, we expand the investigated domain to 2D and analyze the performance of the improved representation. To this end, we set up a 2D study which includes the four different setups *naive*, *position*, *miller*, and *miller and position* as described





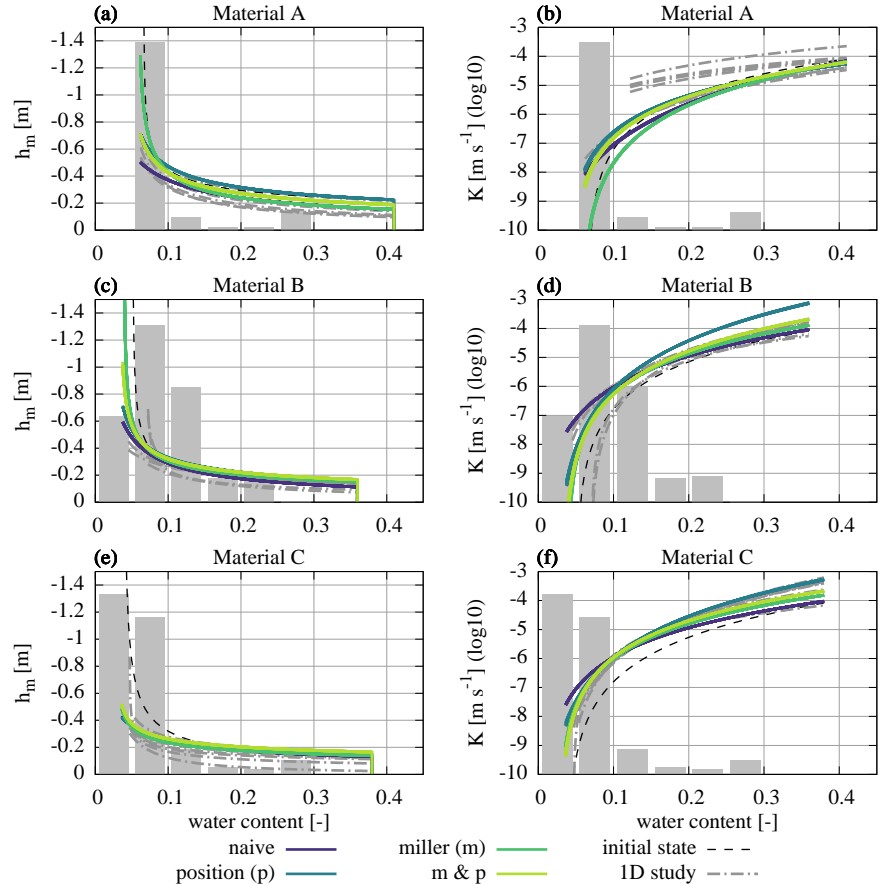

**Figure 13.** We show the resulting material functions for all three materials involved in the 2D study which is analyzed with four setups (*naive*, *position*, *miller*, and *miller and position*). The plot range is adjusted to the available water content range for each material. The line width of the 2D inversion results corresponds to two times the formal standard deviation of the hydraulic parameters. The height of the histogram bars denotes the number of available water content measurements and is normalized over all figures. Since the inversions for all setups are initialized with the material functions resulting from the ***initial state estimation*** (Sect. 3.5.4), the difference between the results is directly linked to the estimation of sensor positions and small-scale heterogeneities. For direct comparison, the results of the *1D study* are also visualized.

above[11]. Since the position of both the tensiometer and the groundwater well is within in the modeled domain, we use the hydraulic potential measurement data as well as the TDR measurement data in this study. Thus, the *position* setup is adjusted, such that both the positions of TDR sensors and the tensiometer are estimated.

Considering that the ensemble members of the 1D study converged reasonably close to the initial state material functions, the

---

[11]Some TDR sensors are located close to or even below the groundwater table. Therefore, the position and the Miller scaling factor could not be estimated for all TDR sensors. No position was estimated for sensors 7, 8, 14, 15, 16, 23, 24, 31, and 32. No Miller scaling factor was estimated for sensors 8, 14, 15, 16, 23, 24, 31, and 32.





inversions for the 2D study are directly initialized with these parameters. The 2D simulations in this work are calculated on a grid with $100 \times 100$ grid cells covering $19.1 \text{ m} \times 1.9 \text{ m}$. The limit of the Newton solver is set to $10^{-8}$ (Sect. 3.5.1). Like for the 1D studies, we choose 0.007 as the standard deviation of the TDR measurements. The standard deviation of the tensiometer (0.025 m) is assessed from the accuracy ($\pm 5$ hPa) as specified by the manufacturer[12]. Lacking an independent estimate for the accuracy of the manual groundwater table position measurement, we employ the accuracy of material interfaces in ASSESS

(Sect. 3.5.3). Same as for the tensiometer, this leads to a standard deviation of 0.025 m.

For the 2D study, the number of sensors is comparable to the number of hydraulic material parameters. Therefore, estimating sensor positions and Miller scaling factors increases the total number of parameters and thus the computational cost considerably: The *miller and position* setup has more than 3 times the number of parameters[13] of the *naive* setup.

The total number of evaluated TDR sensors[14] increased to 25. Hence, the measurement data cover more complicated flow phenomena compared to the 1D studies. Therefore, we expect that (i) the resulting parameters are more reliable and (ii) the description of the measurement data is worse for 2D compared to 1D. The standardized residuals visualized in Fig. 12 confirm this expectation and demonstrate that – similar to the 1D study – representing sensor position and small-scale heterogeneity uncertainty improves the description of the TDR data. Still, the probability plot (Fig. 12b) displays a characteristic *S*-shaped

curve for the TDR data highlighting persisting large residuals during transient phases. The statistical measures given in Table 6 reveal that the $e_{\text{MA}}$ of the *naive* setup merely increases by less than a factor of 2 compared to the *case III* of the 1D study and that estimating sensor positions and Miller scaling factors improves the description of the TDR measurement data by more than a factor of 2 leading to a $e_{\text{MA}}$ of 0.0034.

The description of the hydraulic potential measurement data, however, does exclusively improve in those setups, in which the

sensor position is estimated and the general structure of the model-mismatch does not change significantly. Due to the large input flux during the experiment, a correct representation of the manually measured groundwater table data is impossible in 2D. As soon as the quasi-equilibrium between the well and the site is exceeded, the forcing via the groundwater well instantiates a 3D water flux. The pressure difference between the tensiometer and the groundwater table in the well (Fig. 3) shows that the site is not in quasi-equilibrium during the forcing. Thus, we expect that the simulation predicts a higher position of the ground-

water table in the well during imbibition phases and a lower groundwater table during the drainage phases. This expectation is confirmed by the standardized residuals shown in Fig. 12. The structural model-data mismatch of the tensiometer data indicates that employing the groundwater table as Dirichlet boundary condition overestimates the forcing in the simulation. Therefore, the simulated hydraulic pressure during the imbibition is larger than the measured one which leads to negative residuals. As expected, this behavior reverses during drainage phases.

Since each setup is started from the same initial material function, the difference between the resulting material properties of the setups (Fig. 13) is a direct consequence of the representation of uncertainties in the sensor position and small-scale

---

[12] In order to transfer the given uniform distribution with range $\pm 5$ hPa $\approx \pm 0.05$ m to a Gaussian distribution, we associate this range with the $2\sigma$ interval of a Gaussian (5 % to 95 %). This leads to an approximate standard deviation of $(0.05 \text{ m} \cdot 2)/4 = 0.025$ m.

[13] Number of estimated parameters for the different setups: *naive*: 17, *position*: 41, *miller*: 42, *miller and position*: 66.

[14] We evaluated 5, 12, 8 TDR sensors for the materials A, B, C.





heterogeneities. A more intuitive understanding can be gained for example by closely investigating the initial state estimation
for material B shown in Fig. 5. The measurement data of the sensors 5, 12, and 29 which are approximately 0.6 m above
groundwater table considerably deviate from the estimated function. In order to cope with this deviation, the least squares fit
for the initial state draws the estimated soil water characteristic to higher water contents. Due to the rigidity of the Brooks-
Corey parameterization, this causes an overestimation of the water content at the position of the sensors 0.8 and 1.4 m above
the groundwater table (sensors 28 and 18). As soon as the uncertainty in sensor position and small-scale heterogeneities are
represented in the model, the outlying measurement data can be described without altering the effective material properties.

The 2D study is based on an increased number of water content measurements, additional hydraulic potential measurements,
and a more complicated flow phenomena compared to the previously discussed 1D study (Sect. 5.1). This improves the ability
of the Levenberg-Marquardt algorithm to separate the estimated uncertain model components. Solely for material A, the setups
show a vertical shift in the soil water characteristic. This can be explained with the relatively low number of water content
measurements monitoring transient phases. Although the number of measurements in the dynamical water content range of
material A is comparable to that of material C, Fig. 4 shows that fewer sensors monitor the transient phases in material A
compared to material C.

Although the uncertainty of the measured grain size distribution (Table 1) is large, the resulting material properties confirm
these measurements to the extent, that material A is the finest of all materials and that the properties of materials B and C are
similar.

In summary, according to the statistical measures, the 2D model performs similarly well as the 1D model in describing the
TDR water content measurement data. However, since the 2D inversions are based on more measurement data covering more
complicated flow phenomena, they yield more consistent and reliable material properties. As each setup is started from the
same initial parameter set, the difference between the resulting material functions directly demonstrates the effect of the ac-
cording unrepresented model errors, namely unrepresented uncertainty in sensor positions and small-scale heterogeneity. The
representation of these uncertain model components improves the description of the TDR water content measurement data by
more than a factor of 2 resulting in a $e_{MA}$ of 0.0034. The remaining structural model-data mismatch for the hydraulic potential
data indicates 3D flow phenomena, which were not considered here.

## 6  Summary

We presented a fluctuating groundwater table experiment in a complicated and effectively 2D architecture (ASSESS), which
was monitored with TDR, GPR, and hydraulic potential measurement devices. This kind of experiment provides high variabil-
ity of the measured water content up to and beyond the validity limits of the Richards equation.

Using inversion methods for parameter estimation, it is key to start the simulations close to the measurement data. Hence,
we employed the Brooks-Corey parameterization to estimate the water content between the TDR sensors. Therefore, we as-
sumed hydraulic equilibrium and approximated the hydraulic potential with the negative distance to the groundwater table.




Subsequently, we associated the approximated hydraulic potential at the position of the TDR sensors with the measured water
content and fitted Brooks-Corey parameters for each material. With the resulting parameters, we calculated an estimate for the
initial water content distribution.

We implemented a structural error analysis which is based on the insight that the structural model-data mismatch indicates
incomplete understanding of reality. We demonstrated that the method can detect significant unrepresented model errors, such
as the inclined architecture of ASSESS.

However, as the sufficient complexity of the model and the measurement data for the required accuracy and precision are
unknown a priori, we analyzed the effect of unrepresented model errors by implementing different setups of the 1D and 2D
studies with increasing model complexity. In these setups, the model complexity was gradually increased starting with the
estimation of effective hydraulic material properties and adding the estimation of sensor positions, small-scale heterogeneity,
or both.

In order to investigate, whether the soil water movement at ASSESS can be described with a 1D model, we created three
cases with increasing number of sensors per material located at distinct positions in ASSESS. For each case, we generated an
ensemble of Latin-Hypercube sampled initial parameters for the Levenberg-Marquardt algorithm. Since the resulting material
properties of the best inversions are reasonably close to the parameters estimated for the initial water content distribution, these
may also be used as initial parameters for gradient-based optimization algorithms. We found that with an increasing number
of sensors per material, the structural model-data mismatch increased for those setups, in which only the effective material
properties were estimated. Representing sensor position uncertainty and small-scale heterogeneities, however, improved the
description of the measurement data significantly in the cases with more than one sensor per material. We showed that also due
to unrepresented lateral flow, the resulting material properties of 1D cases are likely to be biased.

Since all setups of the 2D study were initialized with the parameters estimated for the initial water content distribution, the
difference between the resulting material functions show the quantitative effect of the according unrepresented model errors.
Representing sensor position uncertainty and small-scale heterogeneities improved the description of the water content data
significantly, as this decreased the associated $e_{\mathrm{MA}}$ by more than a factor of 2 to 0.0034.

Since the three approaches (i) initial state estimation, (ii) 1D inversion, and (iii) 2D inversion yield similar effective hydraulic
material parameters, we finally discuss their levels of improving the quantitative understanding of soil water dynamics.

The initial state estimation requires at least three water content measurements per material over the full water content range
and the position of the groundwater table to estimate the parameters for soil water characteristic for one specific equilibrated
hydraulic state. The method does not estimate the other parameters $K_{\mathrm{w,0}}$ and $\tau$ required to model soil water dynamics. Addi-
tionally, it is highly susceptible to uncertainties related to the sensor position and small-scale heterogeneities. Yet, the method
is fast (seconds on a local machine) and suitable to provide initial parameters for gradient-based inversion methods.

The 1D inversions are comparably fast (minutes up to hours on a local machine) and can represent transient states. They allow
to estimate all necessary hydraulic material parameters. However, due to unrepresentable lateral flow, the resulting parameters
are likely to be biased.





The unique characteristics of the 2D inversions (days on a cluster with same number of cores as parameters) is the ability to represent lateral flow phenomena which are typically monitored with a high number of sensors. Hence, the consistency of the representation is implicitly checked. Of the three approaches discussed, this the closest one to reality. Therefore, we expect the

10   most reliable material properties here. Still, unrepresented model errors, such as 3D flow phenomena during strong forcing, may lead to biased resulting parameters.

## 7   Data availability

The underlying measurement data is available at http://ts.iup.uni-heidelberg.de/data/jaumann-roth-2017-hess.zip

*Author contributions.* S. Jaumann designed and conducted the experiment, developed the main ideas, implemented the algorithms, and analyzed the measurement data. K. Roth contributed with guiding discussions. S. Jaumann prepared the manuscript with contributions of both authors.

*Acknowledgements.* We thank Jens S. Buchner for the code to process the ASSESS architecture raw data, Angelika Gassama for technical assistance with respect to ASSESS, and Andreas Dörr for helping to set up a beowulf cluster. Additionally, we thank Hannes H. Bauser, Andreas Dörr, and Patrick Klenk for discussions that improved the quality of the manuscript. We especially thank Patrick Klenk and Elwira Zur for assistance during the experiment.

The authors acknowledge support by the state of Baden-Württemberg through bwHPC and the German Research Foundation (DFG) through

grants INST 35/1134-1 FUGG and RO 1080/12-1.





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





**Table 1.** The grain size distribution in percent by weight displays the different granularity of the materials A, B, and C of ASSESS (G. Schukraft, personal communication, Institute of Geography, Heidelberg University, 2010). Whereas the composition of the materials B and C is similar, material A features a higher percentage of fine sand. Since the mechanical wet analysis is time-consuming and laborious, only material B was sampled twice. Thus, 80 g out of approximately 400 Mg were sampled. Due to rounding, the sum of the values is not always 100.

|  |  | grain size range |  | A | $B_1$ | $B_2$ | C |
|---|---|---|---|---|---|---|---|
| gravel | total | $2 - 63$ mm | [%] | 2 | 5 | 4 | 5 |
| sand | total | $63 - 2000$ µm | [%] | 97 | 96 | 95 | 95 |
|  | coarse | $630 - 2000$ µm | [%] | 10 | 24 | 20 | 17 |
|  | medium | $200 - 630$ µm | [%] | 65 | 64 | 68 | 72 |
|  | fine | $63 - 200$ µm | [%] | 22 | 8 | 7 | 6 |
| silt | total | $2 - 63$ µm | [%] | 0 | 0 | 0 | 0 |
| clay | total | $< 2$ µm | [%] | 0 | 0 | 0 | 0 |

**Table 2.** During the experiment, ASSESS was forced with a fluctuating groundwater table. Therefore, 17.8 m$^3$ of water were pumped in and 14.7 m$^3$ were pumped out of the groundwater well. For the calculation of the according flux, the surface area of ASSESS was approximated with 80 m$^2$. All times are given in UTC.

| time start | time end | duration [min] | water volume [m$^3$] | flux [$10^{-6}$ m s$^{-1}$] |
|---|---|---|---|---|
| 12:55:00 | 13:20:00 | 25 | $-0.7649$ | $-6.4$ |
| 14:20:00 | 18:50:00 | 270 | $8.3900$ | $6.4$ |
| 20:35:00 | 23:10:00 | 155 | $4.7809$ | $6.4$ |
| 07:25:00 | 09:55:00 | 150 | $4.6361$ | $6.4$ |
| 12:35:00 | 14:00:00 | 85 | $-3.9970$ | $-9.8$ |
| 15:00:00 | 16:10:00 | 70 | $-3.1709$ | $-9.4$ |
| 16:40:00 | 19:15:00 | 155 | $-6.7299$ | $-9.0$ |





**Table 3.** This overview includes specification whether the considered model error is represented and explicitly estimated within the scope of this study.

| model error | represented | estimated |
|---|:---:|:---:|
| local non-equilibrium | ✗ | ✗ |
| hysteresis | ✗ | ✗ |
| numerical error | ✗ | ✗ |
| orientation of ASSESS | ✓ | ✗ |
| initial state | ✓ | ✗ |
| entrapped air | ✓ | ✗ |
| boundary condition | ✓ | ✓ |
| sensor position | ✓ | ✓ |
| small-scale heterogeneity | ✓ | ✓ |
| material properties | ✓ | ✓ |

**Table 4.** The 1D study comprises three different cases which investigate the three materials with increasing number of TDR sensors per material at different locations in ASSESS (Fig. 1). Note that each material is covered twice.

| case | sensors | materials | position [m] |
|:---:|---|---|---|
| I | 1 & 2 | C, A | 16.16 |
| II | 10, 11 & 12, 13 | C, B | 10.95 |
| III | 25, 25, 27 & 28, 29, 30 | A, B | 1.26 |





**Table 5.** In order to analyze the results of the 1D study, the performance of the best ensemble members for each case and for each setup are benchmarked with statistical measures. With increasing numbers of included TDR sensors per material, the statistical measures for the *naive* setup indicate worse description of the measurement data. However, estimating the position and the Miller scaling factor for each TDR sensor, improves description of the measurement data significantly according to the statistical measures.

| case | setup | | $e_{\mathrm{RMS}}$ | $e_{\mathrm{MA}}$ | $e_{\mathrm{NS}}$ |
|------|---------|-----|--------|--------|------|
| I | naive | | 0.0043 | 0.0033 | 1.00 |
| I | position | (p) | 0.0037 | 0.0028 | 1.00 |
| I | miller | (m) | 0.0045 | 0.0035 | 1.00 |
| I | m & p | | 0.0037 | 0.0028 | 1.00 |
| II | naive | | 0.0067 | 0.0034 | 0.96 |
| II | position | (p) | 0.0053 | 0.0030 | 0.98 |
| II | miller | (m) | 0.0042 | 0.0027 | 0.99 |
| II | m & p | | 0.0042 | 0.0029 | 0.99 |
| III | naive | | 0.0090 | 0.0056 | 0.96 |
| III | position | (p) | 0.0062 | 0.0040 | 0.98 |
| III | miller | (m) | 0.0054 | 0.0031 | 0.98 |
| III | m & p | | 0.0043 | 0.0023 | 0.99 |

**Table 6.** For each setup of the 2D study, the results are benchmarked with statistical measures. Similar to the 1D study, estimating the sensor position and the Miller scaling factors improves the statistical measures related to the water content significantly. The statistical measures for the hydraulic potential which describe both the tensiometer and the groundwater well data improve only for setups in which the sensor positions are estimated.

| setup | | water content | | | hydraulic potential | | |
|--------|-----|--------|--------|------|--------|--------|------|
| | | $e_{\mathrm{RMS}}$ | $e_{\mathrm{MA}}$ | $e_{\mathrm{NS}}$ | $e_{\mathrm{RMS}}$ | $e_{\mathrm{MA}}$ | $e_{\mathrm{NS}}$ |
| naive | | 0.0156 | 0.0099 | 0.92 | 0.036 | 0.030 | 0.99 |
| position | (p) | 0.0098 | 0.0063 | 0.97 | 0.028 | 0.023 | 0.99 |
| miller | (m) | 0.0073 | 0.0047 | 0.98 | 0.036 | 0.031 | 0.99 |
| m & p | | 0.0059 | 0.0034 | 0.99 | 0.022 | 0.018 | 1.00 |