# Peer review of "Unrepresented model errors - effect on estimated soil hydraulic material properties"

_Hydrology and Earth System Sciences, 2017_

## Referee Comment (RC2)

Review of manuscript HESS-2017-109

**Unrepresented model errors –
Effect on estimated soil hydraulic material properties**

Dear Editor:

The study is interesting and demonstrates a huge work. However, before it can be transferred to the HESS step of the journal, I suggest the authors should discuss some key points and possibly make some changes in the text. I apologize for having been a bit late with my appraisal, but this also gave me the opportunity to read the comments from another referee and one discussant.

I have listed below one general comment and several specific remarks, the most significant of which are starred (*).

**General comment**
As a referee, but also as a reader of studies dealing, among various sources of uncertainties, also with those associated with the locations of sensors that monitor a flow process, there is always something causing me some concern. When setting up an experimental test, efforts are made reducing errors (especially the systematic errors) and, among other things, one measures the positions of the various sensors as accurately as possible. I also understand that this task can be a bit more complicated under field conditions, especially when inserting the sensors at the greatest soil depths. Therefore and to the benefit of a wider readership, the authors should justify more why they are interested in this type of uncertainty.

Moreover, I have the feeling that the error in sensor location should be viewed more as a systematic error rather than a random error. I think that the method employed by the authors might not be adequate to treat the presence of systematic errors. Some clarifications and a discussion on this point seem deserving.

**Specific remarks**

- (*) P.1, L.13. The authors claim that the approximated soil water retention function is "reasonable close to the inversion results". Actually and allowing for the types of water flow processes investigated, it would have been more interesting and effective that the favorable outcome is in terms of the unsaturated hydraulic conductivity function. From the results depicted in the right plots of Fig.10 and Fig.13, this does not seem the case.

- P.1, L.20-23. On the topic of inverse modeling applied to Soil Hydrology, I suggest citing the more recent and comprehensive papers by Hopmans et al. (2002) and/or by Vrugt and Dane (2006). Concerning the lab-scale experiment, the paper by Romano and Santini (1999) also treat types of errors of interest for the present study. As for the in-situ applications, the paper by Romano (1993) can also be in line with some aspects of the present study.

- P.1, L.22. The paper by Schneider et al. (2006) was published in HESS, not in Hess-D.

- (*) P.2, L.10-13. It is not clear (at least to me) which processes the authors are talking about. For example, the sensor position is definitely not a process. Moreover, as far as I am aware, the previous studies refer to minimum unknown parameters to be estimated mainly because they employed the classic $\chi^2$ penalty criterion coupled with the Levenberg-Marquardt (LM) algorithm. Why do not compare the present results with those ones whether you use, for example, the DREAM tool developed by Vrugt (2016)? By doing that way, the paper would be even more interesting since the authors claim of having developed a modified LM algorithm.

- (*) P.4, L.8-10. Strictly speaking, the $\theta$-based Richards equation describes the variations in space ($x$, $y$, and $z$ coordinates) and time ($t$) of the volumetric soil water content. Then, due to the selected relationship between water content and matric pressure head, one can retrieve the corresponding variations in $h$.

- (*) P.19, L.25-27. This is a quite common outcome when modeling of data with a maximum likelihood estimator and optimization techniques. I think that this problem should be addressed in another way. Namely, more in terms of the information content of the available input datasets. Does the initial information content increase when adding the additional data? Are the additional data not at all, or weakly, or strongly correlated among them and with the already available input datasets?

- As general and final comment, I should say that the English usage is very good. Nevertheless, the text is hard to follow. I do not have suggestions on this point, but the authors should make any effort to improve this aspect of the manuscript. Also, sub-section 4.1 might be left out from the manuscript, whereas I do not see the need to have so many small sub-sections in Section 3. Section 6, albeit being a summary, seems pointless and ineffective, chiefly because it also contains many repetitions. A real concluding remark section would be more effective, if necessary. Footnotes are rare or even absent in our scientific literature.

**References cited**

Hopmans, J.W., J. Šimunek, N. Romano, and W. Durner, 2002. Simultaneous determination of water transmission and retention properties: Inverse methods. In "Methods of Soil Analysis, Part 4, Physical Methods" (J.H. Dane and G.C. Topp, eds.), pp. 963-1008, SSSA Book Series N.5, Madison, WI, USA, ISBN 0-89118-841-X.

Romano, N. 1993. Use of an inverse method and geostatistics to estimate soil hydraulic conductivity for spatial variability analysis. *Geoderma* 60(1/4):169-186.

Romano, N., and A. Santini, 1999. Determining soil hydraulic functions from evaporation experiments by a parameter estimation approach: Experimental verifications and numerical studies. *Water Resour. Res.* 35:3343-3359.

Vrugt, J.A. 2016. Markov chain Monte Carlo simulation using the DREAM software package: Theory, concepts, and MATLAB Implementation. *Environmental Modeling & Software* 75:273-316.

Vrugt, J.A., and J.H. Dane, 2006. Inverse modeling of soil hydraulic properties. Encyclopedia of Hydrological Sciences 6:77.

---

## Referee Comment (RC1) · Anonymous Referee #1 · 30 Mar 2017

This paper deals with the use of inverse modelling of soil water content and soil pressure measurements for estimating effective hydraulic parameters. Data are obtained from the ASSESS test site, which is an advanced experimental facility with well-known but complex soil layering and well-controlled boundary conditions. In particular, the effect of unrepresented model errors is investigated, and more importantly procedures are proposed to account for these model errors within the inversion process. The representation errors that are considered include uncertain sensor positions, uncertainty in boundary conditions, local heterogeneity, and dimensionality of the model (here: 1D vs. 2D). For the selected boundary condition (multi-step drainage and imbibition from below), it is found that small representation errors in sensor position can significantly affect the inverted material properties.

I am strongly supportive of the idea of this study. Many studies typically stop after a

single inverse modelling run. Sometimes the residuals are inspected, but very rarely the results of inverse modelling are used to improve the model concept or the system representation. This study explores several representation errors, and the results seem to indicate that reasonably small changes in system representation can significantly improve the data fit and the properties of the residuals. However, I have a few general concerns and specific comments that I would like to see addressed. Addressing these comments likely involves moderate to major revisions. In addition, grammar and spelling should be improved in the revised version.

GENERAL COMMENTS 1. The introduction is rather unambitious and does do full justice to the content of the manuscript. The authors decided to include a second introduction in section 4.3 where the structural error analysis is introduced. I strongly encourage bringing the idea of structural error analysis in the beginning of the manuscript to better prepare the reader for what is coming. The general stance of this extended introduction could be: Analysis of inverse modelling results to improve models. As already indicated above, I think there are too few studies that pursue this idea.

2. A general concern with the chosen approach is that the same data are used for inverse modelling and evaluation of the results. Would it not be much stronger when the inversely estimated parameters are tested on an independent dataset? Are such independent datasets available for the ASSESS test site? In the current manuscript, improvements in data fit are reported, but this is fully expected because the amount of parameters was increased at the same time.

3. A short discussion about the transferability of the results to other soil types would also be useful for the readers. Of course, gradients in water content are steep in ASSESS and this may significantly impact the importance of accurate sensor positioning. Would the same insights be obtained when the ASSESS test would have consisted of different loam soils? Please comment.

4. The authors decided to not take the classical structure of Introduction, Materials and

Methods, Results and Discussion, Conclusions. For me, the alternative structure is not really working. For example, part of the results are presented in section 4.3 where the used methods have not yet been clearly explained. Although I may be purist in this matter, I would say that this paper would benefit from an organization following the classical scheme.

SPECIFIC COMMENTS Page 1, Line 1. Abstract should be a single paragraph. In addition, it is customary to provide the scope of the manuscript with an opening statement. Here, the authors immediately jump to the aims of the study.

Page 1, Line 19. Is direct determination really expensive? I would prefer time-consuming here.

Page 2, Line 19. Huisman et al. (2010) considered a soil layer on top of the dike material.

Page 2, Line 21. I would like to see more information about the TDR system that was used. Did the authors rely on automatic waveform analysis, or was this done manually to obtain more accurate results?

Page 2, footnotes. I find it very unusual that the authors use footnotes. Is this possible and common in HESS? In any case, it seemed to me that much of the information provided in the footnotes could have easily been integrated in the main text. Please reduce the amount of footnotes to a minimum.

Page 4, Line 14. One-sentence paragraphs should be avoided.

Page 6, Line 19. I am not so convinced that a separate section on the implementation is a good idea. In particular, I do not really like the three very short subsections that now follow. It makes the text unpleasant to read.

Page 9, Line 5. I could not follow your implementation of small-scale heterogeneity. Are you using heterogeneous parameters fields throughout the domain, or is this heterogeneity only introduced locally? Please clarify.

Page 10, line 12. I know this as global-local approach.

Page 10, line 21. Not sure that standard deviation is appropriate here? Is this not the expected standard deviation of the residuals (e.g. measurement error).

Figure 7. This figure did not make things clearer for me. Consider deleting.

Page 12, Line 5. The start of this section seems out of place. For me, this clearly belongs to the general introduction (see general comments).

Page 13, Line 20-32. Perhaps I am a purist, but for me this is a result and this is not a good position in the paper to discuss a result. I would bring this later.

Figure 9. It would be good to show measured and modelled data in at least one figure. Here, a third column could be added to the left in addition to the residuals.

Page 15, Line 5. Avoid repetitions. This has already been described four lines ago.

Figure 10. This figure is too complicated. I am not sure how to read it. I am particularly unsure about the green.

Page 19, Line 32. It is not so clear how you reached this conclusion. Perhaps this needs to be emphasized better when discussing the results.
* * *

---

## Short Comment (SC1) · 9 Apr 2017

I congratulate the authors to an interesting study at the ASSESS experimental site. I consider the topic and the discussion manuscript highly relevant and worth to be published in HESS. Because of this, I would like to contribute some comments for a revision.

1. If I understand correctly, the authors argue for a retention-dynamics-based identification of soil hydraulic material properties based on inverse modelling of an imbibition and outflow experiment. There have been many studies on the issue of inverse parameter estimation, which I consider relevant for the MS. This also holds for the discussion of heterogeneity and "unrepresented model errors". I.e. the authors name the "validity limits of the Richards equation" but I do not see the conceptual basis of the argumentation for their approach. Moreover, I suggest to present an independent reference for the found parameters (e.g. from laboratory analysis) and to include a critical view on the TDR inferred soil moisture values.

2. Despite my appreciation of the logical intention of the structure of the MS, I find it very difficult to follow. Especially, I could not trace answers to my expectations from the title and abstract – probably because they became obscured by many detailed side-tracks and because some promised elements (like GPR data or elaboration on what are model errors) are not really followed. Maybe a fundamental revision and exhibition of the main story line could clarify most of the forthcoming points.

3. What is the reason to use own models, solvers and the LM least squares optimizer instead of established and tested toolboxes? Is it really matter of the MS to present the technical details and equations although they are not developed further, taken up or discussed later on? How can be assured that numerical errors in the code do not bias the results (see also Clark and Kavetski 2010, 10.1029/2009WR008896)? I can imagine that the details suit well as appendix and that an explanation of the concept and intention to use these tools can clarify much of my second concern.

4. Since heterogeneity is also an issue of scale and conceptual deficiency, I find the arguments not yet well drawn. What support of the TDR sensors is integrated by the measurements? How exactly are the estimated positions of the TDR sensors calculated and how precisely are the real positions known?

5. Since GPR data of the experiment appears to be existing (Klenk et al. 2015 under review in HESSD doi:10.5194/hessd-12-12215-2015) I do not understand, why it is not used for the study (although mentioned in the abstract and introduction)? I suppose that the TDR and GPR data could be a very valuable pair of observations to be compared directly (as both rely on the rel. electrical permittivity). The strong advantage of GPR as spatially continuous technique could be related to the local measurement with higher absolute precision of the TDRs.

6. Figures 10 and 13 suggest to me, that the observations relate to the portion of the (sandy) retention curve which is rather linear (and that the strongly non-linear part is actually only of importance at low matric potential). How is a transfer of the found parameters to the full retention spectrum validated? Since the ASSESS site is an artificial, well-defined test bed I would assume that the actual retention properties are known and that local deviations are mainly due to differences in bulk density. Hence I could imagine that the authors could use fig. 11 in the methods section to explain their approach in much more detail and related to specific research hypotheses referring to the retention properties. At the moment, I find it very difficult to read figure 9 and 12 and to compare the 1D and 2D case.

Please find minor comments highlighted in the attached MS file. All the best, Conrad

Please also note the supplement to this comment:
http://www.hydrol-earth-syst-sci-discuss.net/hess-2017-109/hess-2017-109-SC1-supplement.pdf

**Supplement:**

[revised manuscript text omitted]

---

## Author Comment (AC1) · 8 Jun 2017

We thank the reviewers and Conrad Jackisch for their constructive comments! We revised the manuscript accordingly and refer to it in the replies to the comments.

Please note that the original version of the manuscript included an error in the water content data preprocessing routine. This routine did not associate TDR sensor 27 with material A but with material B. Therefore, measurement data of this sensor which are close to saturation were neglected. We corrected this error and reran the 2D study. Hence, the resulting statistical measures and parameters changed slightly in the revised manuscript.

[Figure]

Please also note the supplement to this comment:
http://www.hydrol-earth-syst-sci-discuss.net/hess-2017-109/hess-2017-109-AC1-supplement.pdf

**Supplement:**

[revised manuscript text omitted]

---

## Author Response (AR1)

**Author's response**

This authors response includes the

- reply to referee comment 1

- reply to referee comment 2

- reply to short comment 1

- change list of the most important changes

- comment concerning the marked up version of the manuscript

- marked up version of the manuscript

Since the replies were uploaded, minor improvements were added to the manuscript and to the replies. Particularly, the line numbers of the references in the included replies were adjusted to keep this response consistent.

**Reply to referee comment 1**

*This paper deals with the use of inverse modelling of soil water content and soil pressure measurements for estimating effective hydraulic parameters. Data are obtained from the ASSESS test site, which is an advanced experimental facility with well-known but complex soil layering and well-controlled boundary conditions. In particular, the effect of unrepresented model errors is investigated, and more importantly procedures are proposed to account for these model errors within the inversion process. The representation errors that are considered include uncertain sensor positions, uncertainty in boundary conditions, local heterogeneity, and dimensionality of the model (here: 1D vs. 2D). For the selected boundary condition (multi-step drainage and imbibition from below), it is found that small representation errors in sensor position can significantly affect the inverted material properties. I am strongly supportive of the idea of this study. Many studies typically stop after a single inverse modelling run. Sometimes the residuals are inspected, but very rarely the results of inverse modelling are used to improve the model concept or the system representation. This study explores several representation errors, and the results seem to indicate that reasonably small changes in system representation can significantly improve the data fit and the properties of the residuals. However, I have a few general concerns and specific comments that I would like to see addressed. Addressing these comments likely involves moderate to major revisions. In addition, grammar and spelling should be improved in the revised version.*

**Reply:** We thank the reviewer for the constructive comments and suggestions. The manuscript was revised accordingly, hence we refer to the revised manuscript.

**General Comments**

*1. The introduction is rather unambitious and does do full justice to the content of the manuscript. The authors decided to include a second introduction in section 4.3 where the structural error analysis is introduced. I strongly encourage bringing the idea of structural error analysis in the beginning of the manuscript to better prepare the reader for what is coming. The general stance of this extended introduction could be: Analysis of inverse modelling results to improve models. As already indicated above, I think there are too few studies that pursue this idea.*

**Reply:** We agree and revised the introduction accordingly.

*2. A general concern with the chosen approach is that the same data are used for inverse modelling and evaluation of the results. Would it not be much stronger when the*

*inversely estimated parameters are tested on an independent dataset? Are such independent datasets available for the ASSESS test site? In the current manuscript, improvements in data fit are reported, but this is fully expected because the amount of parameters was increased at the same time.*

**Reply:** Independent datasets can either be achieved by changing the measurement method or the experiment setup. The former leads to different model errors, e.g., due to different measurement volumes of different instruments, and the latter leads to a different sensitivity of the data on the estimated parameters. Hence, we decided to analyze datasets of different instruments separately and to compare the results. Please also note the reply to comment 5 of SC1.

The improvements are reported so that the readers can judge whether the size of improvement is worth the associated additional effort.

*3. A short discussion about the transferability of the results to other soil types would also be useful for the readers. Of course, gradients in water content are steep in ASSESS and this may significantly impact the importance of accurate sensor positioning. Would the same insights be obtained when the ASSESS test would have consisted of different loam soils? Please comment.*

**Reply:** We agree, that this is an interesting question. Beyond general comments, we cannot answer it with the given data of the presented case study.

*4. The authors decided to not take the classical structure of Introduction, Materials and Methods, Results and Discussion, Conclusions. For me, the alternative structure is not really working. For example, part of the results are presented in section 4.3 where the used methods have not yet been clearly explained. Although I may be purist in this matter, I would say that this paper would benefit from an organization following the classical scheme.*

**Reply:** We revised the structure of the manuscript, bringing it closer to the classical scheme.

**Specific Comments**

*Page 1, Line 1. Abstract should be a single paragraph. In addition, it is customary to provide the scope of the manuscript with an opening statement. Here, the authors immediately jump to the aims of the study.*

**Reply:** We revised the abstract and added an introductory sentence.

*Page 1, Line 19. Is direct determination really expensive? I would prefer time-consuming here.*

**Reply:** We changed the wording here.

*Page 2, Line 19. Huisman et al. (2010) considered a soil layer on top of the dike material.*

**Reply:** We checked the paper again and found that the dike consists of the investigated material (Fig. 2, 4, 6, 9, and 10).

*Page 2, Line 21. I would like to see more information about the TDR system that was used. Did the authors rely on automatic waveform analysis, or was this done manually to obtain more accurate results?*
**Reply:** We added this information in Sect. 2.1 (Page 3, Line 20), Sect. 2.2.4 (Page 6, Line 2), and Sect. A1.3 (Page 23, Line 1).

*Page 2, footnotes. I find it very unusual that the authors use footnotes. Is this possible and common in HESS? In any case, it seemed to me that much of the information provided in the footnotes could have easily been integrated in the main text. Please reduce the amount of footnotes to a minimum.*
**Reply:** We integrated the footnotes in the text.

*Page 4, Line 14. One-sentence paragraphs should be avoided.*
**Reply:** We improved the section, such that the one-sentence paragraph is avoided.

*Page 6, Line 19. I am not so convinced that a separate section on the implementation is a good idea. In particular, I do not really like the three very short subsections that now follow. It makes the text unpleasant to read.*
**Reply:** We decided to separate the more general theory from the case dependent implementation such that the readers can skip or flip through the more general theory and just read the details on the implementation and do not have to do the sorting themselves. The three short subsections were introduced for precise referencing.

*Page 9, Line 5. I could not follow your implementation of small-scale heterogeneity. Are you using heterogeneous parameters fields throughout the domain, or is this heterogeneity only introduced locally? Please clarify.*
**Reply:** We clarified the Sect. A1.4 (Page 23, Line 18).

*Page 10, line 12. I know this as global-local approach.*
**Reply:** We updated the description of the 1D setup (Sect. 2.4.1) and don't use the wording anymore.

*Page 10, line 21. Not sure that standard deviation is appropriate here? Is this not the expected standard deviation of the residuals (e.g. measurement error).*
**Reply:** We made the sentence more precise (Sect. 2.3.1, Page 8, Line 14).

*Figure 7. This figure did not make things clearer for me. Consider deleting.*
**Reply:** We still think that graphically representing the flow of information is useful.

*Page 12, Line 5. The start of this section seems out of place. For me, this clearly belongs to the general introduction (see general comments).*
**Reply:** We revised the introduction accordingly.

*Page 13, Line 20-32. Perhaps I am a purist, but for me this is a result and this is not a good position in the paper to discuss a result. I would bring this later.*
**Reply:** This is intended as an example to show that the method works. It is thus a methods piece, not a result.

*Figure 9. It would be good to show measured and modelled data in at least one figure. Here, a third column could be added to the left in addition to the residuals.*
**Reply:** We added the results of the *miller and position* setup from the 2D case study to the data in Fig. 4.

*Page 15, Line 5. Avoid repetitions. This has already been described four lines ago.*
**Reply:** This comment is unclear to us. We rechecked the paragraph and could not identify any repetition.

*Figure 10. This figure is too complicated. I am not sure how to read it. I am particularly unsure about the green.*
**Reply:** We removed the indication of the setups in order to simplify the figure.

*Page 19, Line 32. It is not so clear how you reached this conclusion. Perhaps this needs to be emphasized better when discussing the results.*
**Reply:** We separated the Sect. 3 in subsections and clarified the analysis in Sect. 3.1.3.

**Reply to referee comment 2**

*Dear Editor:*

*The study is interesting and demonstrates a huge work. However, before it can be transferred to the HESS step of the journal, I suggest the authors should discuss some key points and possibly make some changes in the text. I apologize for having been a bit late with my appraisal, but this also gave me the opportunity to read the comments from another referee and one discussant. I have listed below one general comment and several specific remarks, the most significant of which are starred (\*).*

**Reply:** We thank the reviewer for the constructive comments and suggestions. The manuscript was revised accordingly. Hence, we refer to the revised manuscript.

**General Comments**

*As a referee, but also as a reader of studies dealing, among various sources of uncertainties, also with those associated with the locations of sensors that monitor a flow process, there is always something causing me some concern. When setting up an experimental test, efforts are made reducing errors (especially the systematic errors) and, among other things, one measures the positions of the various sensors as accurately as possible. I also understand that this task can be a bit more complicated under field conditions, especially when inserting the sensors at the greatest soil depths. Therefore and to the benefit of a wider readership, the authors should justify more why they are interested in this type of uncertainty. Moreover, I have the feeling that the error in sensor location should be viewed more as a systematic error rather than a random error. I think that the method employed by the authors might not be adequate to treat the presence of systematic errors. Some clarifications and a discussion on this point seem deserving.*

**Reply:** We agree, that efforts are made to measure the positions of the various sensors as accurately as possible. Yet, the surface and/or the subsurface structure may change with time and requirements for accuracy and precision may change a posteriori. We clarified this in Sect. A1.4 (Page 23, Line 9).

We agree that the uncertainty in the sensor position is a systematic or structural error. This is the reason why this uncertainty was represented and the parameter estimation algorithm was used to propose more consistent positions of the sensors minimizing this systematic error.

**Specific remarks**

*(\*) P.1, L.13. The authors claim that the approximated soil water retention function is*

reasonable close to the inversion results. *Actually and allowing for the types of water flow processes investigated, it would have been more interesting and effective that the favorable outcome is in terms of the unsaturated hydraulic conductivity function. From the results depicted in the right plots of Fig.10 and Fig.13, this does not seem the case.*
**Reply:** Lacking direct measurements of the unsaturated hydraulic conductivity at the position of the TDR sensors, the presented method merely yields an estimate of the initial hydraulic state and an approximation of the soil water characteristic. The remaining parameters for the initial hydraulic conductivity function ($K_s$ and $\tau$) are taken from Carsel and Parrish (1988, 10.1029/WR024i005p00755) and are independent of the presented measurement data. Hence, the presented method is not applicable to approximate the hydraulic conductivity function.

*P.1, L.20-23. On the topic of inverse modeling applied to Soil Hydrology, I suggest citing the more recent and comprehensive papers by Hopmans et al. (2002) and/or by Vrugt and Dane (2006). Concerning the lab-scale experiment, the paper by Romano and Santini (1999) also treat types of errors of interest for the present study. As for the in-situ applications, the paper by Romano (1993) can also be in line with some aspects of the present study.*
**Reply:** We revised the introduction accordingly. Please also note the reply to comment 1 of SC1.

*P.1, L.22. The paper by Schneider et al. (2006) was published in HESS, not in Hess-D.*
**Reply:** We corrected the reference.

*(\*) P.2, L.10-13. It is not clear (at least to me) which processes the authors are talking about. For example, the sensor position is definitely not a process. Moreover, as far as I am aware, the previous studies refer to minimum unknown parameters to be estimated mainly because they employed the classic $\chi^2$ penalty criterion coupled with the Levenberg-Marquardt (LM) algorithm. Why do not compare the present results with those ones whether you use, for example, the DREAM tool developed by Vrugt (2016)? By doing that way, the paper would be even more interesting since the authors claim of having developed a modified LM algorithm.*
**Reply:** We agree and changed the formulation (Page 2, Lines 9 – 12).
As the major focus of the manuscript, we investigate the effect of neglected structural errors which lead to suboptimal results using the $\chi^2$ penalty criterion. Therefore, we also use the $\chi^2$ penalty criterion coupled with the Levenberg-Marquardt algorithm and quantify the effect of unrepresented model errors by resulting residuals and material properties of the different setups (Sect. 2.3 and Sect. 2.4).
In order to compare the best result of the different setups, we are rather interested in maximum likelihood instead of its distribution in this work. The former is more efficiently found with the Levenberg-Marquardt compared to the DREAM algorithm. Additionally, if the $\chi^2$ is used as likelihood function in DREAM, the discussed problem of neglected processes and uncertainties will remain the same as we use a flat prior in this study. Also, adding additional material would make the already long manuscript

even longer.

(*) P.4, L.8-10. Strictly speaking, the $\theta$-based Richards equation describes the variations in space (x, y, and z coordinates) and time (t) of the volumetric soil water content. Then, due to the selected relationship between water content and matric pressure head, one can retrieve the corresponding variations in h.
**Reply:** We changed the wording in Sect. 2.2.1 (Page 4, Line 9).

(*) P.19, L.25-27. This is a quite common outcome when modeling of data with a maximum likelihood estimator and optimization techniques. I think that this problem should be addressed in another way. Namely, more in terms of the information content of the available input datasets. Does the initial information content increase when adding the additional data? Are the additional data not at all, or weakly, or strongly correlated among them and with the already available input datasets?
**Reply:** If the sensors monitor hydraulic dynamics which is not represented perfectly in the model, the residual will increase as the probability to monitor these model errors is increased with the number of sensors. In information theory, the information content of data is often quantified with measures such as the Shannon entropy. In order to apply these measures, the input data have to be transferred to random data. This requires knowledge about the general data structure which has to be gained from the data themselves. This implies massive practical issues in heterogeneous media. Since the TDR data monitor the same process at different positions, the Pearson correlation coefficient of the data is mainly positive and depends in particular on the recorded hydraulic dynamics. As the materials A and C which are flipped in case I and III, the characteristics of the monitored hydraulic dynamics changes. Hence, the correlation of these data is weak in general. The hydraulic state of material B is monitored at a similar position in cases II and III. Thus, the correlation of the according data increases.

As general and final comment, I should say that the English usage is very good. Nevertheless, the text is hard to follow. I do not have suggestions on this point, but the authors should make any effort to improve this aspect of the manuscript. Also, sub-section 4.1 might be left out from the manuscript, whereas I do not see the need to have so many small sub-sections in Section 3. Section 6, albeit being a summary, seems pointless and ineffective, chiefly because it also contains many repetitions. A real concluding remark section would be more effective, if necessary. Footnotes are rare or even absent in our scientific literature.
**Reply:** We revised the general structure of the manuscript. Please note the reply to comment regarding Page 6, Line 19 of RC1. We also revised Sect. 3 and Sect. 4 to make them more concise and generally integrated the footnotes into the text.

[Figure]

Figure 1: Pearson correlation coefficient for the data used in the 1D study

**Reply to short comment 1**

*I congratulate the authors to an interesting study at the ASSESS experimental site. I consider the topic and the discussion manuscript highly relevant and worth to be published in HESS. Because of this, I would like to contribute some comments for a revision.*
**Reply:** We thank Conrad Jackisch for the constructive comments and suggestions. The manuscript was revised accordingly. Hence, we refer to the revised manuscript.

*1. If I understand correctly, the authors argue for a retention-dynamics-based identification of soil hydraulic material properties based on inverse modelling of an imbibition and outflow experiment. There have been many studies on the issue of inverse parameter estimation, which I consider relevant for the MS. This also holds for the discussion of heterogeneity and* unrepresented model errors. *I.e. the authors name the* validity limits of the Richards equation *but I do not see the conceptual basis of the argumentation for their approach. Moreover, I suggest to present an independent reference for the found parameters (e.g. from laboratory analysis) and to include a critical view on the TDR inferred soil moisture values.*
**Reply:** The manuscript combines two main lines of thoughts: One is concerned with the estimation of hydraulic material properties on the basis of TDR measurement data acquired in a complicated subsurface architecture, which was forced with a fluctuating water table. We agree that this approach is not new and was applied already for many one-dimensional systems in the laboratory and also some in the field. The introduction cannot list all of the available literature. Rather it connects to the related literature we deem most relevant for this manuscript. The other line of thought considers a general problem in modeling, namely the investigation, which physical processes and uncertainties have to be represented in order to describe the measurement data adequately. As the true behavior of the system of interest is unknown and since required adequacy depends on the application at hand, we choose to test different hypotheses (realized by increasingly complicated models) and analyze their results. We improved the introduction to better reflect these two lines.
The Richards equation is only valid where water and air phase decouple, i.e. at intermediate saturation. At high saturation, water– and air–flow become coupled and a two–phase formulation is required. Conversely, at low saturation, vapor transport in the air phase is no more negligible and at least a two–components model is required. Richards equation is a single–phase model.

The transfer of laboratory data to field situations is notoriously difficult. Major challenges are (i) bringing an undisturbed sample into the laboratory, (ii) representing structures that are larger than the sample. In our opinion, there thus cannot be such a thing as an independent reference for a field site.

We assessed the precision from TDR data close to saturation and the accuracy with error propagation considering uncertainties in porosity and in bulk permittivity (Jaumann, 2012) yielding an uncertainty of 0.007 volumetric water content (Sect. A2.1). This result is of the same order as the evaluation of Roth et al., (1990, 10.1029/WR026i010p02267). A major point of critique of the Complex Refractive Index Model (CRIM) concerns that it is a physically-motivated and not a physically-based model (e.g., Brovelli and Cassiani, 2008, 10.1111/j.1365-2478.2008.00724.x). Additionally, other uncertainties such as the influence of the electrical conductivity on the evaluated water content and on the temperature model for the permittivity of water as well as the spatial distribution of the relative permittivity of the soil bulk are neglected in the manuscript.

*2. Despite my appreciation of the logical intention of the structure of the MS, I find it very difficult to follow. Especially, I could not trace answers to my expectations from the title and abstract – probably because they became obscured by many detailed side-tracks and because some promised elements (like GPR data or elaboration on what are model errors) are not really followed. Maybe a fundamental revision and exhibition of the main story line could clarify most of the forthcoming points.*
**Reply:** We revised the structure of the manuscript accordingly.

*3. What is the reason to use own models, solvers and the LM least squares optimizer instead of established and tested toolboxes? Is it really matter of the MS to present the technical details and equations although they are not developed further, taken up or discussed later on? How can be assured that numerical errors in the code do not bias the results (see also Clark and Kavetski 2010, 10.1029/2009WR008896)? I can imagine that the details suit well as appendix and that an explanation of the concept and intention to use these tools can clarify much of my second concern.*
**Reply:** The solver for the Richards equation (muPhi) is tested, published (Ippisch et al., 2006, 10.1016/j.advwatres.2005.12.011), and it is, to the best of our knowledge, the numerically most efficient solver. The Levenberg-Marquardt algorithm was implemented according to published literature, because some of the required approaches are not implemented in available toolboxes.

We present only those technical details in the manuscript that are necessary to understand the evaluation procedure, such that the methods are traceable and reconstructible. The position of the methods section depends on the philosophy of the journal.

Due to the discretization of the problem in space and time, numerical errors are always existent, essentially balancing computational effort and numerical accuracy. We chose the grid resolution and meta–parameters given in the manuscript based on a grid convergence analysis.

We adjusted the structure of the manuscript accordingly.

*4. Since heterogeneity is also an issue of scale and conceptual deficiency, I find the arguments not yet well drawn. What support of the TDR sensors is integrated by the measurements? How exactly are the estimated positions of the TDR sensors calculated and how precisely are the real positions known?*

**Reply:** We clarified this issue in Sect. A1.4.

The support of the TDR sensors depends in particular on the sensor design and can be calculated (Robinson, 2003, 10.2136/vzj2003.4440). For the TDR sensors in ASSESS, the measurement volume contains a cylinder with a radius of approximately half the rod distance around the central rod in homogeneous electrical permittivity distribution. The rod distance of the TDR sensors used in ASSESS is 0.03 m.

*5. Since GPR data of the experiment appears to be existing (Klenk et al. 2015 under review in HESSD doi:10.5194/hessd-12-12215-2015) I do not understand, why it is not used for the study (although mentioned in the abstract and introduction)? I suppose that the TDR and GPR data could be a very valuable pair of observations to be compared directly (as both rely on the rel. electrical permittivity). The strong advantage of GPR as spatially continuous technique could be related to the local measurement with higher absolute precision of the TDRs.*

**Reply:** Three single channel time–lapse GPR radargrams were acquired during the experiment and are currently evaluated for a separate publication. The measurement data presented in Klenk et al. (2015) were recorded during a different imbibition and drainage experiment. The main focus of this manuscript is to quantify the effect of unrepresented model errors on the soil hydraulic material properties and to find consistent description of TDR measurement data. These data are characterized by a point–scale measurement volume of the sensors, which is the main reason for the described effect of uncertainties concerning the sensor position and small-scale heterogeneity. Since such point–measurements are rather the rule then the exception in most large–scale studies, the related issues require critical consideration. A rather complementary analysis is required for the GPR data taking into account the larger measurement volume and GPR related representation errors. This would blow the limits of a single paper. Please also note the reply to comment 2 of RC1.

*6. Figures 10 and 13 suggest to me, that the observations relate to the portion of the (sandy) retention curve which is rather linear (and that the strongly non-linear part is actually only of importance at low matric potential). How is a transfer of the found parameters to the full retention spectrum validated? Since the ASSESS site is an artificial, well-defined test bed I would assume that the actual retention properties are known and that local deviations are mainly due to differences in bulk density. Hence I could imagine that the authors could use fig. 11 in the methods section to explain their approach in much more detail and related to specific research hypotheses referring to the retention properties. At the moment, I find it very difficult to read figure 9 and 12 and to compare the 1D and 2D case.*

**Reply:** A transfer of the results to the full retention spectrum can neither be made nor validated with the available water content data and missing hydraulic potential measurements. We explained in the reply to comment 1, why no laboratory–based reference retention properties are known for ASSESS. We think that Fig. 9 is required for the discussion of the results and is best understood with a direct reference to the application. We improved the description, how to read these figures in Sect. 2.3.3 (Page 10, Line 17). Please also note the reply to the comment of RC1 concerning Fig. 7 (Fig. 9 in the unrevised manuscript).

*Please find minor comments highlighted in the attached MS file.*
**Reply:** We revised the manuscript considering these comments.

**List of most important changes**

(Reference to revised version)

- Major revision of the text

- Footnotes were included in the text

- Introduction: Clarified storyline and perspective

- Restructured Section 2:
  Moved application-dependent details of the representation to the appendix A1

- Figure 2: Added face color to indicate 1D studies

- Figure 4: Added simulation of the 2D *miller & position* setup

- Restructured Section 2.3: Structural error analysis
  (includes old Section *Parameter estimation*)

- Restructured Section 3.1: 1D study: Discuss results in subsubsections

- Restructured Section 3.2: 2D study: Discuss results in subsubsections

- Figures 10 and 11 changed slightly due fixed error in data preprocessing

- Added Section 6: Competing interests

- Added Section A1.3: Evaluation of TDR traces

- Added Section A2: Setup

- Tables 5 and 6: Decreased number of significant digits

- Added Table 7: Parameters of the 2D *miller & position* setup

**Comment to the marked up version of the manuscript**

As the manuscript was revised completely, the structure and sections changed considerably. Hence, the standard markup tool for latex (latexdiff) has major problems generating a marked up version of the manuscript. Please excuse format errors, broken references to figures, sections, and tables and use the marked up version as a general indicator for changes.

[revised manuscript text omitted]
_\mathrm{w}(h_\mathrm{m})$ with a saturated water content $\theta_\mathrm{w,s}$ $[-]$, a residual water content $\theta_\mathrm{w,r}$ $[-]$, a scaling parameter $h_0$ $[m]$ well (Dagenbach et al., 2013). This parameterization has four parameters: A scaling parameter $h_0$ $[m]$ related to the air entry pressure ($h_0 < 0$ m), the saturated water content $\theta_\mathrm{s}$ $[-]$, the residual water content $\theta_\mathrm{r}$ $[-]$, and a shape parameter $\lambda$ $[-]$ related to the pore size distribution ($\lambda > 0$). In general, $\theta(h_\mathrm{m})$ shows hysteretic behavior (Topp and Miller, 1966). Neglecting hysteresis, this the parameterization may be inverted for $\theta_\mathrm{w,r} \leq \theta_\mathrm{w} \leq \theta_\mathrm{w,s}$, leading to Inserting the Brooks-Corey $\theta_\mathrm{r} \leq \theta \leq \theta_\mathrm{s}$. This leads to

$$h_\mathrm{m}(\theta) = h_0 \left( \frac{\theta - \theta_\mathrm{r}}{\theta_\mathrm{s} - \theta_\mathrm{r}} \right)^{-1/\lambda}.$$ (2)

Inserting the Brooks–Corey parameterization into the hydraulic conductivity model of Mualem (1976) , yields the Mualem-Brooks-Corey parameterization yields the parameterization

$$K(\theta) = K_\mathrm{s} \left( \frac{\theta - \theta_\mathrm{r}}{\theta_\mathrm{s} - \theta_\mathrm{r}} \right)^{\tau + 2 + 2/\lambda}$$ (3)

for the hydraulic conductivity function

which includes where $K_\mathrm{s}$ $[\mathrm{m\ s^{-1}}]$ is the saturated hydraulic conductivity $K_\mathrm{w,0}$ $[\mathrm{
[revised manuscript text omitted]

| sand | total | $63 - 2000\ \mathrm{\mu m}$ | [%] | 97 | 96 | 95 | 95 |
|  | coarse | $630 - 2000\ \mathrm{\mu m}$ | [%] | 10 | 24 | 20 | 17 |
|  | medium | $200 - 630\ \mathrm{\mu m}$ | [%] | 65 | 64 | 68 | 72 |
|  | fine | $63 - 200\ \mathrm{\mu m}$ | [%] | 22 | 8 | 7 | 6 |
| silt | total | $2 - 63\ \mathrm{\mu m}$ | [%] | 0 | 0 | 0 | 0 |
| clay | total | $< 2\ \mathrm{\mu m}$ | [%] | 0 | 0 | 0 | 0 |

**Table 2.** During the experiment, ASSESS was forced with a fluctuating groundwater table. Therefore, $17.8\ \mathrm{m}^3$ $17.8\ \mathrm{m}^3$ of water were pumped in and $14.7\ \mathrm{m}^3$ $14.7\ \mathrm{m}^3$ were pumped out of the groundwater well. For the calculation of the according flux and equivalent height of the water column $\Delta h_{\mathrm{eq}}$, the surface area of ASSESS was approximated with $80\ \mathrm{m}^2$ $80\ \mathrm{m}^2$. All times are given in UTC.

| phase | time start | time end | duration [min] | water volume [$\mathrm{m}^3$] | flux [$10^{-6}\ \mathrm{m\,s}^{-1}$] | $\Delta h_{\mathrm{eq}}$ [m] |
|---|---|---|---|---|---|---|
| initial drainage | 12:55:00 | 13:20:00 | 25 | $-0.7649$ | $-6.4$ | $-0.01$ |
|  | 14:20:00 | 18:50:00 | 270 | $8.3900$ | $6.4$ | $0.10$ |
| multistep imbibition | 20:35:00 | 23:10:00 | 155 | $4.7809$ | $6.4$ | $0.06$ |
|  | 07:25:00 | 09:55:00 | 150 | $4.6361$ | $6.4$ | $0.06$ |
|  | 12:35:00 | 14:00:00 | 85 | $-3.9970$ | $-9.8$ | $-0.05$ |
| multistep drainage | 15:00:00 | 16:10:00 | 70 | $-3.1709$ | $-9.4$ | $-0.04$ |
|  | 16:40:00 | 19:15:00 | 155 | $-6.7299$ | $-9.0$ | $-0.08$ |

**Table 3.** This overview includes specification whether the considered model error is represented and explicitly estimated within the scope of this study.

| model error | represented | estimated |
|---|:---:|:---:|
| local non-equilibrium | ✗ | ✗ |
| hysteresis | ✗ | ✗ |
| numerical error | ✗ | ✗ |
| orientation of ASSESS | ✓ | ✗ |
| initial state | ✓ | ✗ |
| entrapped air | ✓ | ✗ |
| boundary condition | ✓ | ✓ |
| sensor position | ✓ | ✓ |
| small-scale small–scale heterogeneity | ✓ | ✓ |
| material properties | ✓ | ✓ |

**Table 4.** The 1D study comprises three different cases which investigate the three materials with increasing number of TDR sensors per material at different locations in ASSESS (Fig(Fig. 2). ??). Note that each material is covered twice.

| case | sensors | materials | position [m] |
|:---:|---|:---:|---:|
| I | 1 & 2 | C, A | 16.1616.16 |
| II | 10, 11 & 12, 13 | C, B | 10.9510.95 |
| III | 25, 25, 27 & 28, 29, 30 | A, B | 1.261.26 |

**Table 5.** In order to analyze the results of the 1D study, the performance of the best ensemble members for each case and for each setup are benchmarked with statistical measures. With increasing numbers of included TDR sensors per material, the statistical measures for the *naive basic* setup indicate worse description of the measurement data. However, estimating the position and the Miller scaling factor for each TDR sensor, improves description of the measurement data significantly according to the statistical measures.

| case | setup | | $e_{RMS}$ | $e_{MA}$ $e_{NS}$ |
|---|---|---|---|---|
| I | naive basic | | 0.0043 0.004 | 0.0033 1.000.003 |
| I | position | (p) | 0.0037 0.004 | 0.0028 1.000.003 |
| I | miller | (m) | 0.0045 0.005 | 0.0035 1.000.004 |
| I | m & p | | 0.0037 0.004 | 0.0028 1.000.003 |
| II | naive basic | | 0.0067 0.007 | 0.0034 0.960.003 |
| II | position | (p) | 0.0053 0.005 | 0.0030 0.980.003 |
| II | miller | (m) | 0.0042 0.004 | 0.0027 0.990.003 |
| II | m & p | | 0.0042 0.004 | 0.0029 0.990.003 |
| III | naive basic | | 0.0090 0.009 | 0.0056 0.960.006 |
| III | position | (p) | 0.0062 0.006 | 0.0040 0.980.004 |
| III | miller | (m) | 0.0054 0.005 | 0.0031 0.980.003 |
| III | m & p | | 0.0043 0.004 | 0.0023 0.990.002 |

**Table 6.** For each setup of the 2D study, the results are benchmarked with statistical measures. Similar to the 1D study, estimating the sensor position and the Miller scaling factors improves the statistical measures related to the water content significantly. The statistical measures for the hydraulic potential which describe position of the groundwater table including both the tensiometer and the groundwater well data improve only for setups in which the sensor positions are estimated.

| setup | | water content | | | water table | |
|---|---|---|---|---|---|---|
| | | $e_{RMS}$ | $e_{MA}$ | $e_{NS}$ $e_{RMS}$ | $e_{MA}$ | $e_{NS}$ |
| naive basic | | 0.0156 0.017 | 0.0099 0.011 | 0.92 0.04 | 0.036 0.030 | 0.990.03 |
| position | (p) | 0.0098 0.011 | 0.0063 0.006 | 0.97 0.02 | 0.028 0.023 | 0.990.02 |
| miller | (m) | 0.0073 0.008 | 0.0047 0.005 | 0.98 0.03 | 0.036 0.031 | 0.990.03 |
| m & p | | 0.0059 0.006 | 0.0034 0.004 | 0.99 0.02 | 0.022 | 0.02 |

**Table 7.** We present the effective hydraulic material parameters obtained with the setup *miller and position* of the 2D study. The formal standard deviations of the parameter estimation are given with the understanding that these are specific to the applied algorithm and will change for different algorithm parameters. The estimation for the saturated hydraulic conductivity of the gravel layer and for the offset to the Dirichlet boundary condition are $10^{-0.728 \pm 0.006}$ m s$^{-1}$ and $-0.034 \pm 0.001$ m, respectively.

| material | 0.018 $h_0$ [m] | 1.00$\lambda$ [−] | $K_\mathrm{s}$ [m s$^{-1}$] | $\tau$ [−] | $\theta_\mathrm{r}$ [−] | $\theta_\mathrm{s}$ [−] |
|---|---|---|---|---|---|---|
| A | $-0.184 \pm 0.005$ | $1.94 \pm 0.07$ | $10^{-4.212 \pm 0.004}$ | $0.33 \pm 0.07$ | $0.025 \pm 0.004$ | $0.41$ |
| B | $-0.174 \pm 0.004$ | $2.54 \pm 0.06$ | $10^{-3.77 \pm 0.02}$ | $0.78 \pm 0.05$ | $0.035 \pm 0.001$ | $0.36$ |
| C | $-0.159 \pm 0.004$ | $3.28 \pm 0.02$ | $10^{-3.70 \pm 0.02}$ | $0.74 \pm 0.06$ | $0.026 \pm 0.002$ | $0.38$ |